# Chiral superstructures of inorganic nanorods by macroscopic mechanical grinding

Zhiwei Yang[1,2], Yanze Wei [1,2], Jingjing Wei [1] & Zhijie Yang[1] ✉

The development of mechanochemistry substantially expands the traditional synthetic realm at the molecular level. Here, we extend the concept of mechanochemistry from atomic/molecular solids to the nanoparticle solids, and show how the macroscopic grinding is being capable of generating chirality in self-assembled nanorod (NR) assemblies. Specifically, the weak van der Waals interaction is dominated in self-assembled NR assemblies when their surface is coated with aliphatic chains, which can be overwhelmed by a press-and-rotate mechanic force macroscopically. The chiral sign of the NR assemblies can be well-controlled by the rotating directions, where the clockwise and counter-clockwise rotation leads to the positive and negative Cotton effect in circular dichroism and circularly polarized luminescence spectra, respectively. Importantly, we show that the present approach can be applied to NRs of diverse inorganic materials, including CdSe, CdSe/CdS, and $TiO_2$. Equally important, the as-prepared chiral NR assemblies could be served as porous yet robust chiral substrates, which enable to host other molecular materials and induce the chirality transfer from substrate to the molecular system.

The application of mechanical force to the chemical systems, in terms of mechanical grinding or milling, is an attractive means to conduct chemical reactions, which has greatly expanded the synthetic potential and leads to diverse applications including the screening of pharmaceutical co-crystals, solvent-free organic synthesis and discovery of new motifs of inorganic solids[1–5]. Moreover, linking mechanochemistry to supramolecular fields fuels intense studies of emerging supramolecular materials such as coordinating molecular cages, metal-organic frameworks, and interlocked rotaxanes and so forth[6–10]. Driven by multiple interparticle supramolecular interactions, assembly of inorganic nanoparticles (NPs) into diverse ordered superstructures could be a productive means to engineer macroscopic materials with emergent nanoscopic functionalities[11–16]. Nevertheless, one of the canons of self-assembly at the nanoscale is that the physical and/or chemical properties of the NP superstructures are inherently coded in their individual counterpart. One exceptional example is chiral superstructures from achiral NPs with emergent chiroptical properties, which potentially expand the application scope of traditional inorganic solids, exemplified as enantioselective catalysis and optoelectronics[17–21]. Thus, the question emerges: Is it possible to induce the symmetry breaking in well-ordered NP superstructures by exerting mechanical forces, thereby giving rise to chiral NPs assemblies from achiral NPs?

Bridging the achiral building blocks with the chiral NP superstructures requires the induction of spatial asymmetry. The use of chiral templates has been demonstrated to be a simple yet productive strategy to render the NP assemblies chiral, and the diversity of chiral templates enables to engineer the rotary optical activity of the nanocomposites[22–27]. Recently, Lu et al. showed that high optical asymmetry is found in helical assemblies of achiral Au nanorods (NRs) with polypeptides, originated from long-range organization of plasmonic NRs[28]. Very recently, we showed that the chiral signs of the self-assembled nanocomposites from achiral NPs and chiral π-conjugated molecules could be modulated by the achiral NPs given by their van der Waals interactions[29,30]. In addition to the chiral templates, chiral assembly of NPs could be implemented by introducing external chiral fields, exemplified as circular polarized light and magnetic field[31–35]. Nevertheless, such strategy requires the responsive ability of NPs to

[1]Key Laboratory of Colloid and Interface Chemistry, Ministry of Education, School of Chemistry and Chemical Engineering, Shandong University, Jinan 250100, PR China. [2]These authors contributed equally: Zhiwei Yang, Yanze Wei. ✉e-mail: zyangchem@sdu.edu.cn

the specific external fields, which only limits to a few highly specific systems. One strategy to tackle this limitation could be based on the mechanical forces macroscopically. Kotov and coworkers demonstrated that twisting a thin film comprised of elastic polymers and plasmonic Au NPs is able to produce reconfigurable chiroptical nanocomposites, which transmits the chirality from the macro- to the nanoscale[36]. Furthermore, layer-by-layer (LBL) assembly of films from one-dimensional nanowires realizes the production of chiral photonic crystals, in which a constant twisting angle persists between the neighboring layer and the nanowires in each layer orient in the same direction[37,38].

Herein we describe chiroptical materials from self-assembled nanorod (NR) superstructures, whose chirality at nanoscale can be controlled via post-assembly modifications using macroscopic grinding. We show that conventional hydrophobic colloidal NRs, regardless of their inorganic composition and aspect ratios, can be equipped with chiroptical features when the self-assembled NR solids are subjected to the mechanical grinding (Fig. 1a). Structural characterizations reveal that the macroscopic mechanical force enables to reorient the NRs within the assemblies due to the weak van der Waals interactions between NRs. We further demonstrate that the assembled chiral NR

superstructures can serve as chiral templates, which successfully modulates the chiroptical response of the nanocomposites.

## Results

### Self-assembly of CdSe/CdS NRs

We first synthesized colloidal NRs of various inorganic compositions, such as CdSe, $TiO_2$, and CdSe/CdS core/shell structures, and their surfaces were capped with aliphatic chains, which render them dispersible in nonpolar solvent (i.e., n-hexane)[39–42]. Most tests in this study were carried out with CdSe/CdS core/shell NRs because their aspect ratios and emissive properties could be facilely tuned by modifying the synthetic approach. Three typical CdSe/CdS NRs with average lengths and diameters of (3.4, 13.9 nm), (3.5, 22.5 nm), and (3.5, 41.7 nm), respectively, were synthesized from the identical CdSe seeds of ~3 nm in diameter, denoted as NR1, NR2, and NR3, respectively (Supplementary Fig. 1). After synthesis, their surfaces were subjected to a ligand exchange process and were coated with dodecanethiol. Self-assembly of CdSe/CdS NRs was carried out by a drop-casting method, where CdSe/CdS NRs dispersed in a mixed solvent of n-hexane/octane (9/1, v/v) were deposited onto a carbon-coated copper grid or quartz substrate at 60 °C. After their self-assembly, transmission electron

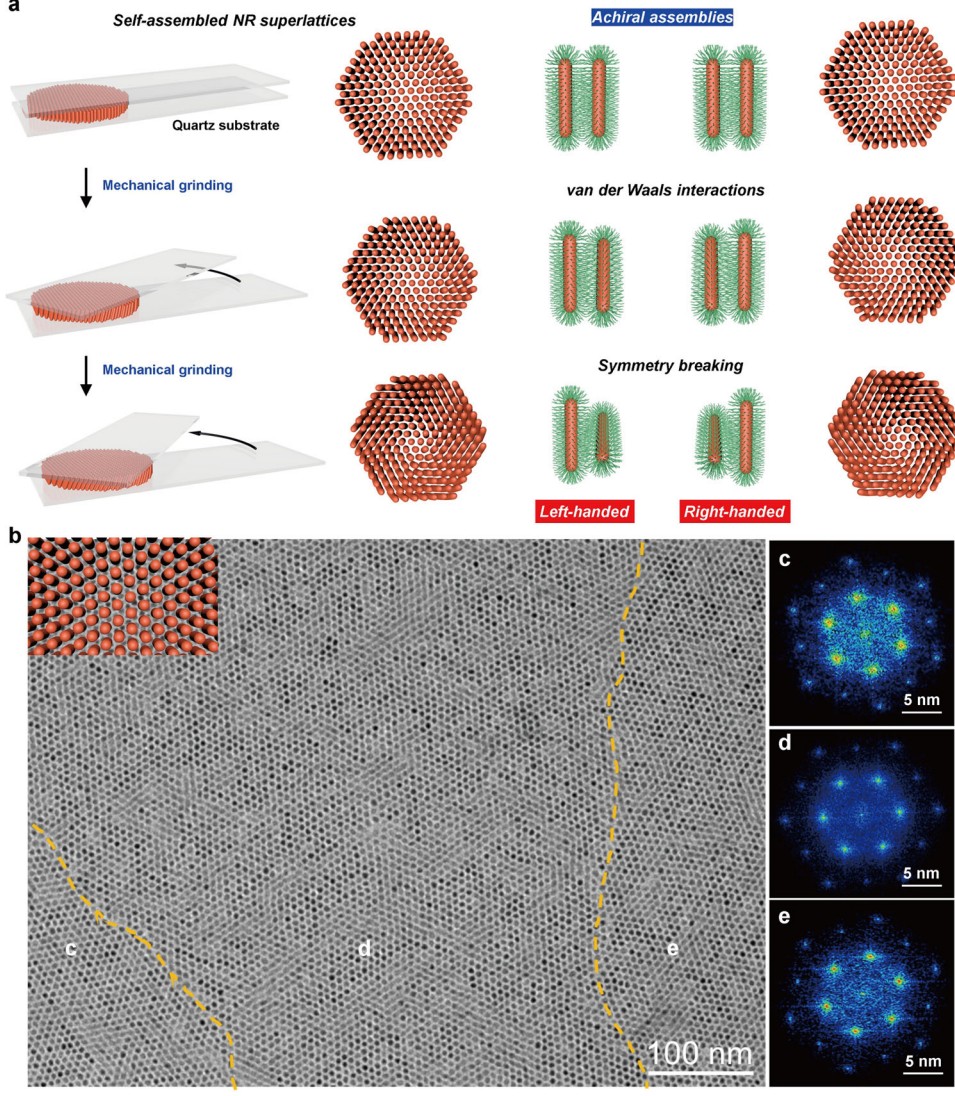

**Fig. 1 | Design principle for exerting chirality of NR assemblies mechanically. a** Schematic illustration of the chirality generation in self-assembled NR superlattices through a press-and-rotate mechanical force. **b** TEM image of self-assembled CdSe/CdS NR hexagonal superlattices vertically on the TEM grid. **c**–**e** Three domains with distinct orientations could be observed and confirmed by the corresponding FFT patterns.

microscopy (TEM) image reveals that nearly all the NR1 were ordered vertically on a substrate, leading to the formation of hexagonal lattice with multiple domains, as confirmed by the corresponding fast Fourier transform (FFT) patterns (Fig. 1b–e). Assembly of elongated NRs results in the oblique packing of the NR2 and NR3 on a substrate (Supplementary Fig. 2). We note that such a drying-mediated self-assembly of NR may generate several distinct orientations, which probably leads to the spatial heterogeneity of the self-assembled thin film[43,44]. Photoluminescence (PL) spectrum of the NR2 assemblies reveals a bathochromic shift (~8 nm) compared to that of the NR2 dispersed in solution (Supplementary Fig. 3). Circular dichroism (CD) spectra reveal that these NR assemblies made from NR1, NR2, or NR3 are CD silent (Supplementary Fig. 4).

## Mechanical grinding of CdSe/CdS NR assemblies

The interparticle attraction between these CdSe/CdS NRs is mainly from the van der Waals attractive forces of either the inorganic core-core or the aliphatic chain-chain interactions, and the latter one is considered to be dominant with the pairwise interaction energy of tens of $k_B$T ($k_B$ and T is the Boltzmann constant and absolute temperature, respectively) or several $10^{-19}$ J[45–49] (The calculation and estimation could be found in Supplementary Note 1). When a pressure ranging from 0.1 to 0.3 pN nm$^{-2}$ (100–300 kPa) was loaded to the NR with a surface area approximately from tens to a few hundred of nm$^2$, we estimate that the applied force to each NR is a few tens of pN, which is capable of moving/rotating nanoscale objects. The work is on the order of $10^{-19}$ J, which is comparable to the attractive energy between two NRs. In this regard, we hypothesize that the structure of these NRs assemblies is transformable when competing external forces are applied, even though they are in densely packed state. To test this hypothesis, we drop casted the NR2 on one quartz substrate (width × length, 10 × 40 mm$^2$ or 10 × 10 mm$^2$), followed by covered another quartz slab of identical size, which was functionalized with a layer of hydrophobic silane (heptadecafluorodecyltrimethoxysilane, FAS). In the first set of experiments, we loaded a pressure of ~200 kPa to the cover slab, followed by moving the cover slab along its long axis back and forth with a speed of ~1 cm s$^{-1}$ for 5 cycles. The as-processed NR2 assemblies were studied by a commercial spectropolarimeter (JASCO J-1500) to understand the CD/linear dichroism (LD) effect and intriguingly, both CD and LD signals could be clearly observed from the respective spectra (Supplementary Fig. 5). We note that the observed CD signal of such sample is rather weak, confirmed by the low dimensionless dissymmetry factor (g-factor), reading $g = \triangle\varepsilon/\varepsilon = 2(\varepsilon_L - \varepsilon_R)/(\varepsilon_L + \varepsilon_R)$ ($\varepsilon_L$ and $\varepsilon_R$ are the absorption coefficients for the respective left and right circularly polarized light) ($g = 0.0025$). Moreover, we have rotated the same sample at various angles, and the LD spectra revealed that nearly mirror spectra could be produced between 0° and 90°, suggesting that LD effect is dominant in this sample. Nevertheless, this preliminary experiment clearly shows that the macroscopic mechanical force could nontrivially reshape the NRs assemblies and render these NRs solids polarizable responding to the light.

To further improve the g-factor of the NR assemblies and mitigate the LD effect, we applied a rotational motion to the cover slab, which consequently induces the torques to each NRs. Specifically, we loaded a pressure of ~200 kPa to the cover slab, and subsequently rotated the slab either clockwise (CW) or counterclockwise (CCW) at an angular velocity of 1 rad s$^{-1}$, akin to the mechanical grinding in mechanochemistry. We first took NR2 assemblies as an example to illustrate the chiral features in the assemblies subjected to the mechanical grinding. Marked positive CD signals with multiple peaks at 426, 480, and 552 nm could be observed for NR2 assemblies under CW grinding (Fig. 2a, b). In sharp contrast, negative CD signals with an almost mirror image could be observed for NR2 assemblies under CCW grinding (Fig. 2b). Noteworthy, the grinding method could simultaneously increase nearly

one order of magnitude the intensity of the CD spectra, but decrease the artefact, LD intensity (Supplementary Fig. 6). To estimate the contribution of LD to the observed CD (CD$_{OBS}$) quantitatively, we applied the semi-empirical equation[50]: CD$_{OBS}$ = CD + LD × 0.02. Therefore, the contribution of LD in the CD$_{OBS}$ in terms of percentage (2LD/CD$_{OBS}$%) is calculated to be ~1.5% (Supplementary Fig. 6 and Supplementary Note 2). Moreover, we have measured the both faces of the quartz slab for NR2(CW) sample, and the almost identical CD signals could be observed (Supplementary Fig. 7), which thereby rules out the effect of linear birefringence (LB). We note that the above technique is a preliminary tool to identify the contribution of LD to the observed CD in a commercial spectropolarimeter, and more advanced techniques, such as Mueller Matrix Polarimetry, could identify the optical chirality with better precision[51,52].

To further confirm the chiral symmetry breaking phenomena in NR assemblies, we have made additional observations as follows: (1) These CD spectra with either positive or negative Cotton effect could be well reproduced under respective CW or CCW grinding, but the intensities of the CD spectra vary from one to the others with a dispersity of ~20%, probably due to the inhomogeneous film thickness of the NR assemblies (Fig. 2c, d). In fact, the film thickness of NR assemblies strongly impacts the observed CD intensities, and a thin film of roughly 300 nm in thickness leads to a rather low CD intensity (Supplementary Fig. 8 and Supplementary Note 3). (2) We have measured the CD spectra at different zones within a same sample NR2-CW (Fig. 2e, f, Supplementary Figs. 9 and 10 and Supplementary Note 4), which display similar CD signals with positive Cotton effect, while the intensities at the maxima of the CD spectra range from 115 to 130 millidegrees (mdeg). (3) Marked changes in CD spectra could be observed for NR assemblies from NRs differing by the lengths (Fig. 2g–i). Namely, the intensity of the CD spectra increases with the increase of the length of the NRs, which could also be interpreted by the dimensionless dissymmetry g-factor, and the g-factor of the NR1, NR2 and NR3 assemblies is $2.6 \times 10^{-3}$, $9.1 \times 10^{-3}$, and $1.18 \times 10^{-2}$, respectively (Fig. 2i). Noteworthy, these CD spectra are highly reproducible as confirmed by the independent measurements of multiple samples and the mirror spectra between the CW and CCW grinding (Supplementary Fig. 11). (4) Furthermore, a control experiment was conducted on the isotropic CdSe/CdS quantum dots (QDs) of ~4.1 nm in diameter, and the results showed that the film of CdSe/CdS QDs does not show any CD signal after the grinding treatment in either CW or CCW direction (Supplementary Fig. 12). Such experiments indicate that the present grinding strategy only applies to the anisotropic NRs but not the isotropic NPs, despite that the surface of the NPs was functionalized with the identical aliphatic chains. (5) Another control experiment was carried out by treating the NR2 assemblies with dithiol (1,10-decanedithiol), which enables to cross-link the NRs within the assemblies covalently. The results showed that no apparent CD signal could be observed after the grinding treatment for the NR2(dithiol), suggesting the pivotal role of the weak van der Waals attractions between the NRs in exerting the chirality of the system.

The effects of various grinding conditions, such as the applied pressure, the rotating speed, as well as the grinding processing time, on the observed CD signals of NR2 assemblies were investigated in a precise manner with a homemade setup shown in Supplementary Fig. 13. In the first set of experiments, we loaded different weights to the samples, which were able to generate pressures of 10, 50, 100, 200, and 300 kPa. All the experiments were repeated three times under CW rotations (1.0 rad s$^{-1}$, 21 s). The results revealed that very low (noisy) CD signals could be produced under 10 kPa. The CD intensity increases with the increase of the loaded pressure, and reaches a maximum value at 200 kPa. Such a trend could be understood by the g-factor value at 490 nm (Supplementary Fig. 14), which increases with the increase of the loaded pressure. This result could be interpreted that the low pressure could not induce the sufficient shear force that is capable of

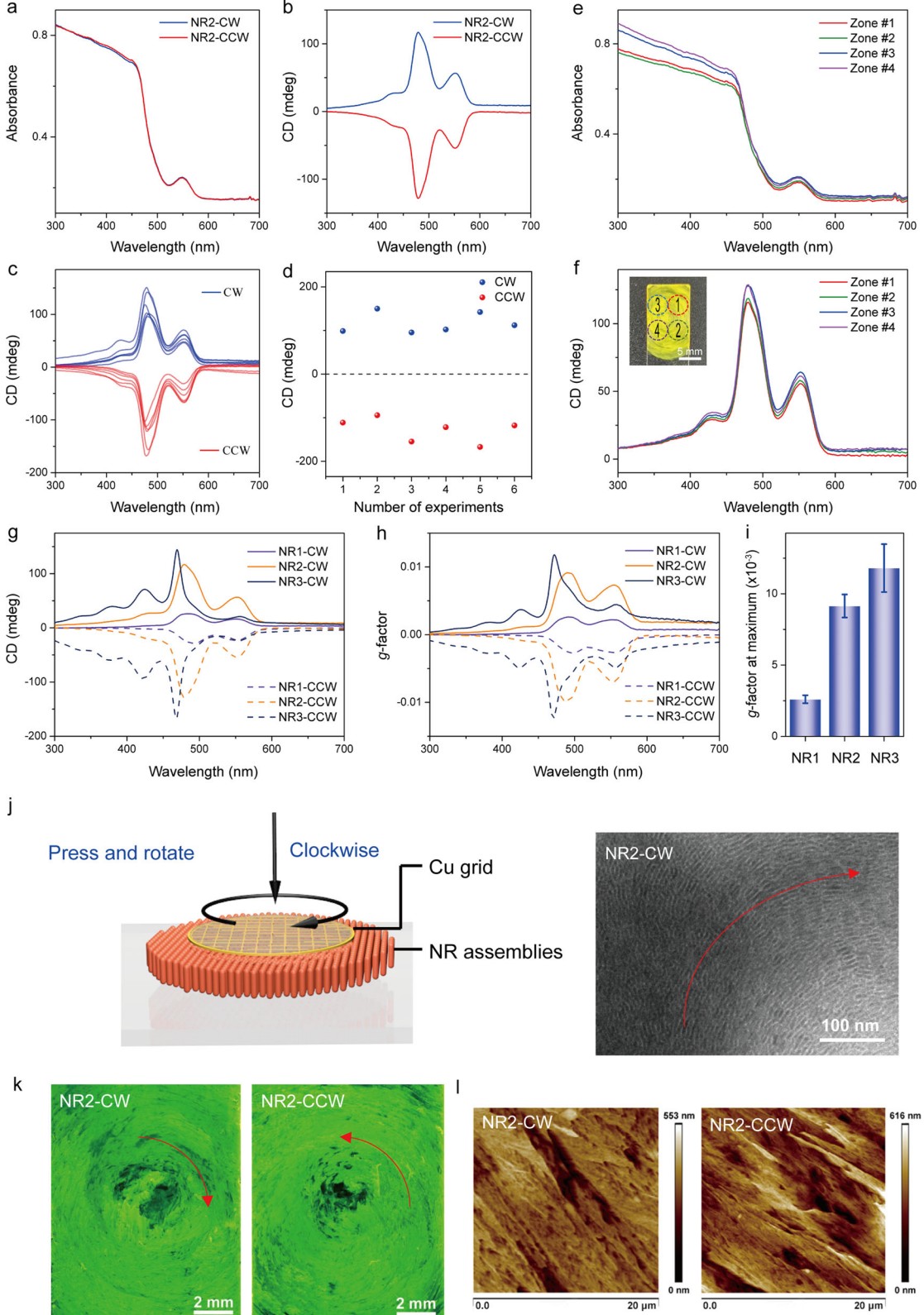

**Fig. 2 | Chiral CdSe-CdS NRs assemblies generated mechanically. a** UV−Vis absorption spectra and **b** CD spectra of NR2-CW (blue) and NR2-CCW (red) assemblies. **c** CD spectra of NR2-CW (blue) and NR2-CCW (red) assemblies produced from independent experiments. **d** CD values at 480 nm for various NR2-CW (blue) and NR2-CCW (red) assemblies from panel (**c**). **e** UV−Vis absorption and **f** CD spectra of NR2-CW assemblies at four distinct regions of a same quartz substrate. CD spectra (**g**) and the corresponding g-factor curves (**h**) of NR1, NR2 and NR3 assemblies after mechanical grinding at either CW or CCW directions. **i** Comparison of the g-factor values of the NR1-CW, NR2-CW and NR3-CW samples. Error bars were determined by standard deviation (SD). **j** Preparation of NR2-CW sample for TEM observation (left) and the TEM image of the NR2-CW sample. **k** Optical images of NR2-CW and NR2-CCW under UV light illumination. **l** AFM images of the NR2-CW and NR2-CCW samples, which showed the magnified area after the grinding process.

breaking the van der Waals interactions between NRs. Another control experiment has been done by keeping the pressure constant (200 kPa), differing by the rotating speed, in terms of angular velocity at 0.1, 0.25, 0.5, 1.0, and 1.5 rad s$^{-1}$ (Supplementary Fig. 15). Likewise, all the experiments were repeated three times under CW rotations for 21 s. We found that CD intensities of the NR assemblies are highly associated with the rotation speed. With the increase of the of the rotation speed, the CD intensity increases, which could also be observed in the g-factor at 490 nm. This result is interesting and is unexpected. In the fluid system, the shear stress increases with the increase of shear rate (rotation speed). However, here the NR assemblies were solids at room temperature, which also showed the "fluid-like" behavior—the increase of the rotation speed (shear rate) leads to the greater shear stress, which in turn results in the higher g-factor of the NR assemblies. The "fluid-like" behavior of the NR assemblies could be associated with their surface aliphatic chains, which equip these NRs with very low intermolecular attractive energy. Lastly, we performed the control experiments by tuning the grinding processing time (Supplementary Fig. 16), while keeping the other conditions constant (pressure of 200 kPa, rotation speed 1.0 rad s$^{-1}$, CW direction). It could be found that a short period of the grinding (7 s) could not lead to the strong CD signals of the NR assemblies, which increases with the increase of the processing time. This can be understood that a short period of rotation could not efficiently drive the breaking of the attractive van der Waals interactions between NRs. After 21 s, strong CD signal could be observed. It is worth noting here that further prolonging the processing time (>21 s) could not result in stronger CD signal, indicating that these NR assemblies were in the stable state.

The structural information of the chiral NR assemblies after grinding was studied by TEM (Fig. 2j). To do so, we placed a carbon-coated Cu grid (400 mesh, Ted Pella) on the top of the NR2 assemblies, followed by pressing and rotating the grid at -100 kPa by using a metallic bar with a flat end of 4 mm in diameter (Note: It requires several attempts not to damage the carbon film). TEM results showed that these NR2 were packed without any translational ordering, which markedly differs from the native NR2 assemblies before grinding. More importantly, circular pattern could be observed, and NR2 are assembled along with the tangent line of the circles, which thereby enables the cross-stacking between the adjacent NRs within the assemblies (Fig. 2j). Such circular pattern of the NR2 assemblies could also be confirmed by the macroscopic images under UV light (Fig. 2k). The circularly packed NRs assemblies could also be observed from the atomic force microscopy (AFM) images (Fig. 2l and Supplementary Fig. 17). These structural features could also be observed for NR1 and NR3 assemblies, and the cross-stacking of NRs is more profound in NR3 assemblies with longer length (Supplementary Fig. 18). However, it is rather difficult to determine the precise configuration of the NRs within the assemblies. Previous report on the dimers of Au NRs showed that the cross-stacking of two NRs through space could induce the chiral interaction between the NRs, which originates from the electronic coupling between the NRs[53–55]. Hence, we conclude that the switching the NR assembly modes from the native paralleled packing to the cross-stacking exerted by the mechanical force is the structural origin for the chiral induction in NR assemblies.

Given the chiral assembling behavior and the highly emissive feature of the NR2 assemblies, we further studied the circularly polarized luminescence (CPL) properties of the NR2 assemblies after grinding. To this end, we synthesized three distinct CdSe/CdS NRs with different emissive bands centered at 540, 571, and 601 nm, respectively, termed as NR2(540), NR2(571), and NR2(601) (Supplementary Fig. 19). We note that these NRs have similar sizes in terms of both diameter and length (Supplementary Fig. 19). The emissive properties of the NRs are mainly regulated by the core size of CdSe, and a larger sized CdSe core would shift the emission band of NR to a longer wavelength. As might be expected, these NR assemblies show intense

CD signals after grinding treatment, and the handedness of the chiral assemblies could be well controlled by the rotation directions (Fig. 3a–c). Likewise, chiral assemblies subjected grinding at CW and CCW directions, termed as NR2(540)-CW and NR2(540)-CCW, lead to mirror-image CPL signals with positive and negative signs, respectively (Fig. 3d, e). The sign of the CPL spectra is in line with the respective CD spectra, and the degree of the CPL could be determined by the luminescence dissymmetry factor ($g_{lum}$), reading $|g_{lum}| = 2(I_L - I_R)/(I_L + I_R)$, where $I_L$ and $I_R$ are the luminescence intensities of left and right circular polarized light, respectively[56]. The $g_{lum}$ for NR2(540)-CW and NR2(540)-CCW is 0.041 and −0.040, respectively. Similarly, the $g_{lum}$ for NR2(571)-CW, NR2(571)-CCW, NR2(601)-CW, and NR2(601)-CCW is 0.040, −0.045, 0.046, and −0.038, respectively (Fig. 3e). We note that such high $g_{lum}$ of the CdSe/CdS NR assemblies is comparable to the data of our recent report on the chiral nanocomposites composing of CdSe/CdS NRs and chiral perylene diimide derivatives (-0.05)[18]. Regarding the facile preparation and the high reproducibility of the present mechanical approach, we envisage that the present chiral films could be a compelling candidate for the CPL-active materials.

## Other chiral inorganic NR assemblies enabled by grinding

Next, we show that the present chiral symmetry-breaking approach could be applicable to other NR assemblies. In this regard, we have synthesized TiO$_2$ and CdSe NRs coated with oleic acid and tetra-decylphosphonic acid, respectively, and these surface ligands would provide the similar van der Waals interactions within in the NR assemblies. The average diameters and lengths of TiO$_2$ and CdSe NRs are (3.1, 25.0 nm) and (2.2, 7.1 nm), respectively (Supplementary Fig. 20), and they were able to self-assemble into ordered structures driven by the chain-chain van der Waals interactions. These NR assemblies did not show any CD signals after their deposition to the quartz substrate (Supplementary Fig. 21). Intriguingly, these NR assemblies showed strong CD signals after CW or CCW grinding treatments (Fig. 4). For TiO$_2$ NR-CW assemblies, CD spectrum shows a broad yet weak band in the visible range with positive Cotton effect, along with two maxima at 365 and 325 nm and one minimum at 350 nm, which are in close association with their intense absorption in the UV range (Fig. 4a–c). As might be expected, the TiO$_2$ NR-CCW assemblies lead to the mirror image to that of the TiO$_2$ NR-CW assemblies. Such bisignate Cotton effect could be well observed in the corresponding g-factor curves, indicating the chiral nature of the assemblies (Fig. 4c). For CdSe NR-CW assemblies, two maxima at 511 and 587 nm could be observed in the CD spectrum, which is attributed to the absorption band at 463 and 567 nm, respectively (Fig. 4d–f). The inversed chirality could also be observed in CdSe NR-CCW assemblies. Repeated experiments at least 10 times suggest that the present approach is reliable to produce chiral assemblies with controlled handedness (Supplementary Fig. 22).

## Chirality transfer from chiral NR assemblies to molecular aggregates

Since grinding the NR assemblies switches the stacking of NRs from side-by-side to the cross-stacking, we hypothesize that the unique chiral assembly of NRs could give rise to "chiral pores" at the nanoscale, which potentially host other guest molecules and thereby enable the chirality transfer from the NR assemblies to the guest molecules (Fig. 5a)[57–59]. To test this hypothesis, we have deposited the molecular dye of Sudan I soluble in ethanol (4 mM, 50 μL) onto the NR2-CW or NR2-CCW assemblies. After the deposition of Sudan I, the UV−Vis absorption spectrum of NR2/Sudan I-CW shows the enhanced absorption in the range from 300 to 500 nm, in which Sudan I has intense absorption (Fig. 5b, c). Interestingly, CD spectrum of NR2/Sudan I-CW shows distinct band compared to that of the native NR2-CW assemblies, where the peak shifts from 483 to 474 nm and from 519 to 513 nm, accompanied by the decrease of the intensity from 102

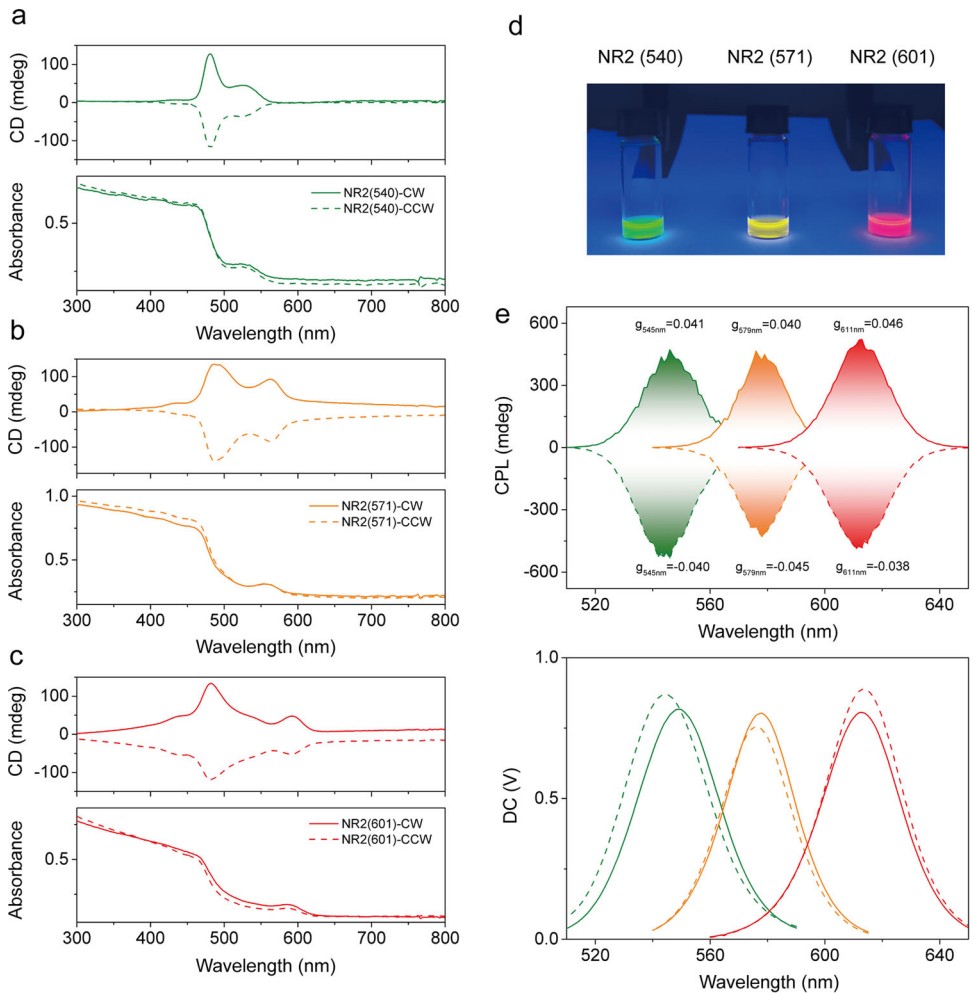

**Fig. 3 | Chiroptical properties of CdSe-CdS NRs with different emissions. a** CD spectra and UV−Vis absorption spectra of NR2(540) assemblies. **b** CD spectra and UV−Vis absorption spectra of NR2(571) assemblies. **c** CD spectra and UV−Vis absorption spectra of NR2(601) assemblies. **d** Photographs of NR2(540), NR2(571), and NR2(601) colloidal solutions under UV light. **e** CPL/PL spectra of the NR2(540), NR2(571), and NR2(601) assemblies.

to 40 mdeg and from 35 to −4 mdeg, respectively (Fig. 5c). Such preliminary data suggest that the chirality transfer takes place from chiral NR assemblies to the molecular assemblies, but their spectral overlapping could be problematic for further analysis. We further used the dye of Sudan blue II that has absorption peaks centered at 596 and 643 nm (Fig. 5d). The CD spectrum of NR2/Sudan blue II-CW shows that two additional bands centered at 593 and 645 nm could be observed, which strongly suggests the chiral nature of molecular assemblies. Similarly, a mirror curve in the CD spectra could be observed when the NR2-CCW assemblies was used as substrate (Fig. 5e).

These dyes could be easily removed through immersing the substrate into the ethanol several times, and the CD spectrum of the sample reveals that no more CD signals could be detected over 590 nm, suggesting that all the dye molecules have been removed completely (Fig. 5f). Importantly, we showed that such dye deposition-removal could be repeated at least 5 times without the apparent decrease of the CD signals from both dyes and NR2-CW assemblies, indicating the robustness of the as-prepared chiral porous host (Fig. 5g, h and Supplementary Figs. 23 and 24). It is important to note that the handedness of the induced chirality of Sudan blue II is opposite to that of the NR2-CW substrate, given by the respective positive and negative Cotton effect at 550 and 645 nm. To further confirm the chirality inversion between the NR2-CW substrate and the deposited dyes, we measured the CPL properties of the

nanocomposites of NR2/Sudan blue II-CW and NR2/Sudan blue II-CW (Fig. 5i). CPL spectrum of NR2/Sudan blue II-CW shows two distinct bands centered at 565 and 680 nm, displaying positive and negative CPL signals, which refer to the emissions from NR2 and Sudan blue II, respectively. The $g_{lum}$ at 565 and 680 nm is determined to be 0.045 and −0.035, respectively, confirming the opposite handedness between the Sudan blue II and the NR2-CW substrate (Fig. 5j). This behavior is also verified by the CPL spectrum of nanocomposite NR2/Sudan blue II-CCW, where negative and positive CPL signals are detected for NR2-CCW substrate and Sudan blue II, respectively. Overall, the robust chiral NR assemblies provide a new productive means to engineer chiral porous materials, in addition to the well-known chiral molecular capsules, chiral mesoporous silica, chiral metal-organic frameworks (MOFs) and chiral covalent-organic frameworks (COFs).

## Discussion

In summary, we have shown how to generate chirality of achiral NR assemblies through a macroscopic mechanical approach. The weak van der Waals interactions in the NR assemblies render these NR assemblies deformable under external force, thereby switching the assembly mode from side-by-side close-packing to the cross-stacking, which consequently leads to the chiral interactions between NRs. The chiral nature of the NR assemblies was validated from diverse inorganic materials, such as CdSe, $TiO_2$, and CdSe/CdS core/shell NRs, and was detected by both CD and CPL spectra. Importantly, these chiral

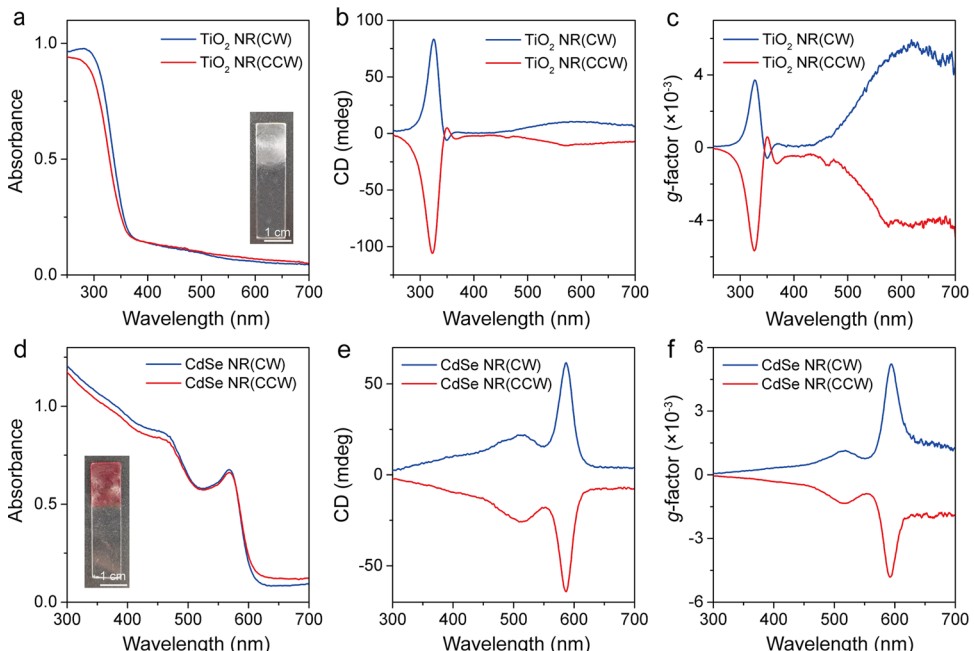

**Fig. 4 | Chiroptical properties of other NR assemblies subjected to mechanical grinding. a–c** UV−Vis absorption spectra, CD spectra and *g*-factor curve of TiO₂ NR assemblies. **d–f** UV−Vis absorption spectra, CD spectra and *g*-factor of CdSe NR assemblies.

assembled NR assemblies could be used as chiral substrate, which enables the chirality transfer from the NR assemblies to the molecular system, akin to the host-guest interactions. Equally important, such chiral substrate is substantially robust and is able to be used for several times without the apparent damage. Overall, the present work expands the mechanochemistry at the molecular level traditionally to the mesoscale level and, more intriguingly, it links the chirality at the nanoscale to the mechanical forces at the macroscale.

## Methods

### Materials

Selenium powder, sulfur, trioctylphosphine (TOP), trioctylphosphine oxide (TOPO), tetradecylphosphonic acid (TDPA), tributyl phosphate (TBP), TiCl₄ and triphenylphosphine (TPP) were purchased from Macklin. Ethanol, hexane, toluene, isopropanol, octadecene (ODE), oleylamine (OAm) and oleic acid (OA) were purchased from Sinopharm. Dodecanethiol (DDT), octane, cadmium oxide, Sudan I and Sudan blue II were purchased from Sigma-Aldrich. All chemicals were used without further purifications.

### Synthesis of CdSe-CdS NRs

Synthesis of CdSe seeds was based on a modified procedure from previous report[39,40]. Typically, CdO (30 mg), octadecyl phosphorous acid (ODPA, 150 mg) and trioctylphosphine oxide (TOPO, 2.0 g) were mixed in a 50 mL four-neck flask, followed by degassing in vacuum at room temperature for 15 min. Then, the temperature was slowly increased to 150 °C for 40 min. After that, the mixture was heated to 300 °C under N₂ flow until it became clear. Subsequently, 1.0 g of trioctylphosphine (TOP) was injected into the mixture at 300 °C followed by increasing the temperature to 365 °C in 10 min. Then 220 µL of the as-prepared TOP-Se solution (1.7 M) was quickly injected into the mixture (TOP-Se: 671 mg of selenium powder was dissolved in 5 mL of TOP under ultrasonication). The heat mantle was quickly removed and cold toluene was added into the colloidal solution to quench the further crystal growth (seed(487) was made by injection at 365 °C followed by rapid quenching; seed(519) and seed(530) were synthesized by injection at 360 °C and 365 °C, respectively, and cooled rapidly; seed(561) was synthesized by injection at 380 °C and cooled naturally).

The as-formed particles were precipitated with ethanol and were subjected to the intensive washing with hexane/ethanol cycles. Finally, these CdSe seeds were dissolved in 5 mL TOP (5 g L⁻¹). The detailed synthetic conditions for various seeds for the NRs of distinct lengths and emission bands were summarized in Supplementary Table 1.

In a typical synthesis of CdSe/CdS NRs, CdO (30 mg), ODPA (150 mg), TOPO (1.0 g) and hexaphosphoric acid (HPA, 40 mg) were mixed in a 50 mL four-neck flask, followed by degassing in vacuum at room temperature for 15 min. Then the temperature was slowly increased to 150 °C for 40 min. Next, the mixture was heated to 300 °C under N₂ flow until it became clear. When the temperature rose to 350 °C, 300 µL TOP-S (60 g L⁻¹) and different content CdSe seeds solutions were injected into the solution to obtain quantum NRs of different lengths (TOP-S: 600 mg sulfur powder was dissolved in 10 mL TOP under ultrasonication). When the temperature dropped to 350 °C, the solution was kept for 8 min followed by the removal of the heating mantle. After the solution was cooled to room temperature, ethanol was added into the solution to precipitate the particles. The CdSe/CdS NRs were further washed with hexane and ethanol to remove the redundant ligands, and a ligand exchange procedure was then performed, by adding dodecanethiol (DDT) into the solutions with the 1.5 times weight as NRs and stirring at 50 °C for 24 h. Finally, the NRs with DDT ligand were purified with ethanol for three times and dissolved in 5 mL hexane for further uses (5 g L⁻¹).

### Synthesis of CdSe NRs

CdSe NRs were synthesized based on a previous report[39,41]. CdSe NRs capped with TOP, TOPO, TDPA, and TBP were synthesized using a Schlenk-line. Typically, CdO (210 mg), TDPA (898 mg), and TOPO (2.26 g) were mixed and heated to 140 °C under vacuum for degassing. After that, the mixture solution was heated to 320 °C to form a clear solution under N₂ purge. Next, the Cd precursor was cooled to room temperature and aged for 12 h under N₂. Then, the Cd precursor was re-heated to 320 °C, while the Se precursor solution (62 mg of Se powder, 1.44 g of TOP and 190 mg of TBP) was quickly injected. The solution was kept for 20 min to obtain CdSe NRs. Finally, the NRs with TDPA (TBP) ligand were washed with ethanol for three times and stored in hexane.

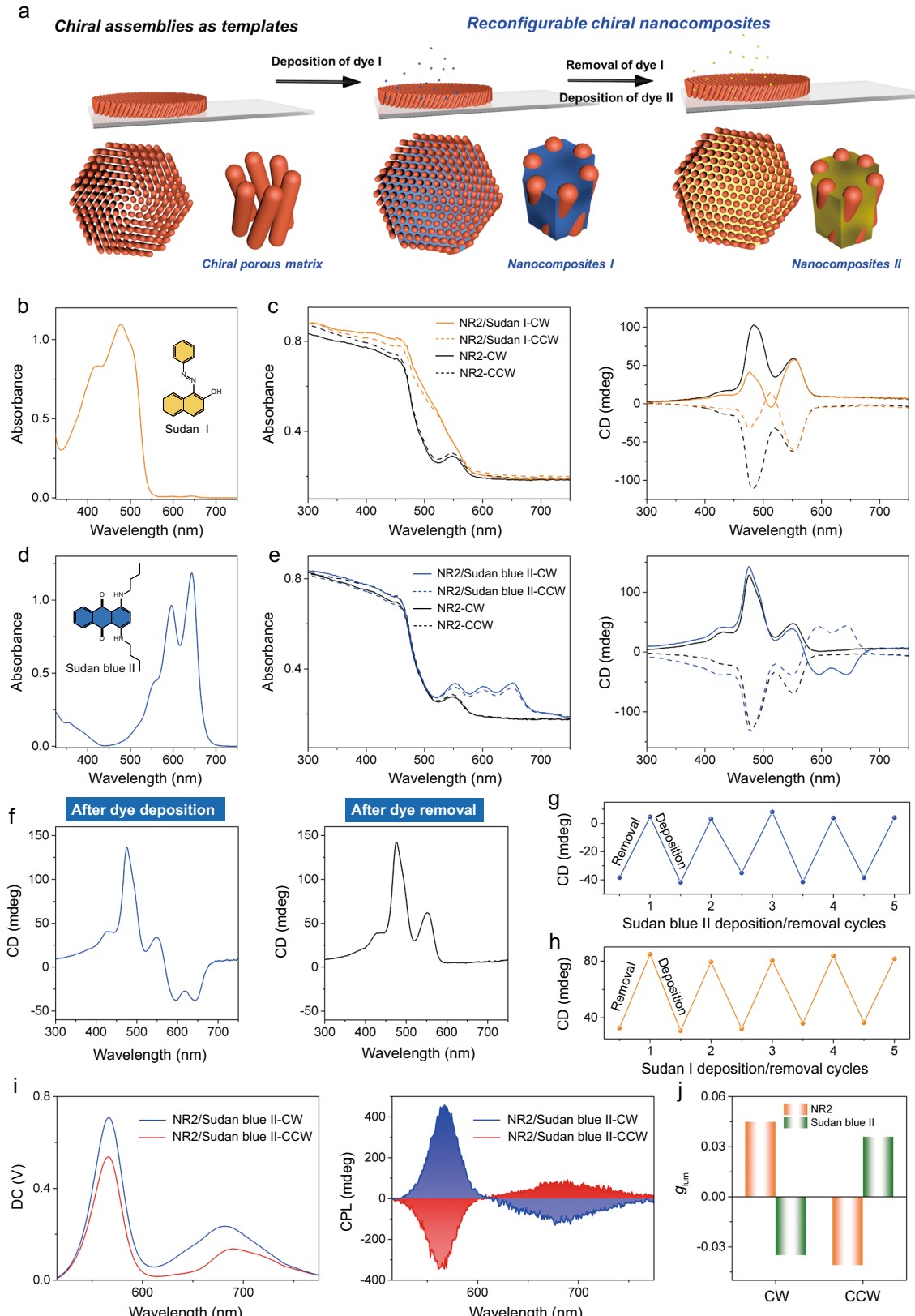

**Fig. 5 | Chirality transfer from the NR assemblies to the molecular systems.**
**a** Scheme of deposition-removal of dye on NR assemblies, which shows chirality transfer from the NR assemblies to the dye molecules. **b** UV−Vis absorption spectrum of Sudan I. **c** UV−Vis absorption spectra and CD spectra of NR2/Sudan I-CW, NR2/Sudan I-CCW, NR2-CW, and NR2-CCW. **d** UV−Vis absorption spectrum of Sudan blue II. **e** UV−Vis absorption spectra and CD spectra of NR2/Sudan blue II-

CW, NR2/Sudan blue II-CCW, NR2-CW, and NR2-CCW. **f** CD spectra of NR2/Sudan blue II-CW after dye deposition and dye removal. **g** CD values at 645 nm of NR2/Sudan blue II-CW after Sudan blue II deposition and removal. **h** CD values at 474 nm of NR2/Sudan I-CW after Sudan I deposition and removal. **i** PL/CPL spectra of the NR2/Sudan blue II-CW and NR2/Sudan blue II-CCW assemblies. **j** Histogram of $g_{lum}$ values for NR2 and Sudan blue II of NR2/Sudan blue II-CW.

## Synthesis of TiO$_2$ NRs

TiO$_2$ NRs were synthesized based on a modification of previous reports[42]. First, ODE (10 mL), OAm (10 mL) and OA (0.48 mL) were heated at 100 °C for one hour under vacuum to remove water and oxygen. Then, the mixture was cooled down to 60 °C under N$_2$ and 1.5 ml of Ti-precursor solution containing 0.2 M TiCl$_4$ and 1.0 M OA in ODE was injected into the solution (dried ODE and OA). The mixture was quickly heated up to 290 °C and held for 10 min. After that, 8 mL of extra Ti-precursor solution was added dropwise into the solution at a rate of 0.25 mL min$^{-1}$. The TiO$_2$ NRs were collected and washed with isopropanol after cooling to room temperature (centrifugation at 7871 g for 8 min). The product was further purified twice by the addition of hexane and isopropanol. The obtained TiO$_2$ NRs were dispersed in n-hexane.

## Substrate treatment

The quartz substrates (40 × 10 mm$^2$ or 10 × 10 mm$^2$) were initially washed in deionized water and ethanol followed by ultrasonication treatment for 30 min. Next, the quartz substrates were silanized by heptadecafluorodecyltrimethoxysilane (FAS) molecule in a reduced pressure environment at room temperature for 24 h and then heating at 80 °C for 3 h. The substrates were washed with ethanol, acetone and isopropanol. Then silanized quartz substrates were stored in hexane before used.

## Sample preparation

Self-assembly of NRs (CdSe/CdS, CdS or TiO$_2$) was carried out through a drop-casting method. In a typical experiment, CdSe/CdS NRs was dispersed in a mixed solvent containing hexane and octane with a volume ratio of 9/1 (5 g L$^{-1}$), followed by their deposition on the quartz substrate (as already cleaned). The solvent was evaporated by placing the substrate on a hot plate of 60 °C, which subsequently triggers the assembly of NRs into superlattices.

The mechanical grinding experiment was carried out by placing a silanized quartz substrate (40 × 10 mm$^2$) on the top the quartz substrate deposited with the NRs assemblies. The two slabs were fixed by a homemade device. A pressure of ~200 kPa was loaded onto the packed quartz slabs with hand. Rotation of the cover slab for 21 s with an annular velocity of 1 rad s$^{-1}$ at either CW or CCW directions.

In order to investigate the various grinding conditions, we have devised a homemade setup shown in Supplementary Fig. 13, which enables us to control the pressure (by various weights), the rotating speed, as well as the mechanical grinding processing time in a precise manner. In the first set of experiments, we loaded different weights to the samples, which were able to generate pressures of 10, 50, 100, 200, and 300 kPa. All the experiments were repeated three times under CW rotations (1.0 rad s$^{-1}$, 21 s). Another control experiment has been done by keeping the pressure constant (200 kPa), differing by the rotating speed, in terms of angular velocity at 0.1, 0.25, 0.5, 1.0, and 1.5 rad s$^{-1}$. Lastly, we performed the control experiments by tuning the grinding processing time, while keeping the other conditions constant (pressure of 200 kPa, rotation speed 1.0 rad s$^{-1}$, CW direction).

## Deposition of dyes onto the NR assemblies

The substrate for the dye deposition experiments was prepared as described in the above section, where NR2-CW or NR2-CCW assemblies were on the quartz substrate. To proceed the dye deposition, the dye molecule (Sudan I or Sudan blue II) was dissolved in ethanol with a concentration of 4 mM, followed by the deposition of 50 μL of the dye solution onto the substrate by a drop-casting method. The carrier solvent (ethanol) was removed by evaporation under 50 °C for 20 min. Finally, the as-prepared NR-dye composites were characterized by CD/CPL. To proceed the dye deposition/removal cycle, the as-deposited NR/dye composites were immersed into ethanol for 5 min at least three times, followed by the dye deposition (4 mM, 50 μL).

## Structural characterizations

UV–Vis absorption spectra were recorded using a UV-1900 spectrophotometer (Shimadzu). Fluorescence spectra were recorded using a FluoroMax-4 (Horiba). CD was characterized by JASCO-1500 (JASCO). CPL spectra were recorded using a ChirascanV100 Circular Dichroism Spectrometer (Applied Photophysics). TEM characterization was performed on a Hitachi HT7700 microscope operating at 100 kV. AFM characterization was performed on a Bioscope Resolve Atomic Force Microscopy (Bruker).

## Reporting summary

Further information on research design is available in the Nature Research Reporting Summary linked to this article.

## Data availability

The data that support the finding of this study are available from the corresponding author upon request.

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

## Acknowledgements

This work was financially supported by the National Natural Science Foundation of China (Grant No. 21972076 and 51903140) and the Natural Science Foundation of Shandong Province (ZR2021JQ05 and ZR2020QB109). Z.Y. thanks Taishan Scholars Program of Shandong Province (tsqn201812011).

## Author contributions

Z.W.Y. and Y.Z.W. performed the experiments. Z.W.Y., Y.Z.W., and J.J.W. discussed and analyzed the data. Z.J.Y. conceived and supervised the project. Z.J.Y. wrote the manuscript and all of the authors commented on to the manuscript.

## Competing interests

The authors declare no competing interests.
