## [Peer Review File · Nature Communications]

Reviewer comments, initial review

Reviewer #1 (Remarks to the Author):

The outlined manuscript describes a novel, mechanical strategy to achieve structurally chiral assemblies of nanorods (NRs) showing chiroptical properties. The method is based on grinding of solid-state nanorod assemblies, which causes cross-stacking of otherwise parallel NRs. The developed strategy has been tested against different composition of semiconductor NRs (CdSe/CdS, CdSe, TiO₂) and different aspect ratio of NRs, attesting that this methodology provides access to materials exhibiting CD and CPL tuned with the type of NR. Moreover, reversible tuning of chiroptical properties was achieved by adding /removing different organic additives into/from NR assemblies. Claims of the Authors are supported with data from different spectroscopic (UV-Vis, PL, CD, CPL) and microscopic (SEM, TEM, AFM) analyzes.

In my opinion this work is truly novel and the proposed methodology well tested using various materials. The described method provides access to circularly polarized luminescent structures which are of great interest for different optoelectronic technologies, and recently attract considerable interest. Thus, I find this publication suitable for publication in Nature communications journal, however only after resolving some issues outlined below.

Major comments:

1. In my opinion speaking of CD properties and providing g-factors for NR₂ assemblies has limited physical meaning and could be misleading. It seems that after averaging all spectra collected for different rotations (Fig. S4) the CD would be 0.
2. The CD is measured using a commercial CD spectrometer which is prone to artifacts when thin films are measured, as noted by the Authors. A first check that can be performed is to measure CD by rotating the sample, which was done here, partially ensuring that artifacts due to linear anisotropy are negligible. The sample was also be flipped, but even then it is unfortunately not enough to be sure that the CD measured is "true" CD. A more advanced technique such as Mueller Matrix Polarimetry or generalized ellipsometry should be used at least on one representative sample to clearly demonstrate this, e.g. A. Mendoza-Galván et al, Mueller matrix spectroscopic ellipsometry study of chiral nanocrystalline cellulose films, 2018 J. Opt. 20 024001; Optical Chirality Determined from Mueller Matrices, Appl. Sci. 2021, 11, 6742. <https://doi.org/10.3390/app11156742>.
3. It is not clear to me where 'circularly packed NRs assemblies' are visible in AFM images - I recommend to highlight these assemblies.
4. I'm not convinced that this method could be described as 'mechanochemical'. Maybe mechanical, or 'mechano-physical' (mechanical force gives rise to chiroptical phenomena). If Authors agree with this comment a partial rewriting of the introduction section would be required.
5. Experimental details should be provided to more precisely describe fabrication of samples doped with dyes. Also, I wonder how true are the simplified schemes showing composites in figure 5. Are the dye molecules actually infiltrating in between NRs? Or rather deposit as aggregates in cracks seen in AFM images?

Minor comments:

6. Wavelength at which CD is measured, shown in panels g and h in Fig. 5, should be clearly stated in the figure and/or figure caption.
7. All schemes of NR assemblies showcase vertical assemblies, whereas oblique assemblies of NR₂ and NR₃ give higher g factor values - maybe it would be worth to include at least one scheme for NR₂/NR₃?
8. Circular/linear dichroism are phenomena, not methods, this sentence should be modified: 'The as-processed NR₂ assemblies were studied by CD/linear dichroism (LD)'
9. What is the role of silanization of the top quartz substrate used for grinding?
10. Authors mention that electronic coupling could be responsible for the emergent chiroptical properties. Thus, maybe it is worth highlighting the difference between photoluminescence of NR samples in dispersions and solid state, confirming the coupling phenomena.
11. I wonder if there shouldn't be a difference in CD /CPL dependent on the distance to the center

of grinded sample?

12. It is surprising to me that TEM grid survived grinding - was this method working smoothly or it required several attempts not to damage the carbon film?

Reviewer #2 (Remarks to the Author):

This work shows chirality in nanorod assemblies can be generated by using macroscopic grinding, which extends scope of mechanochemistry to macroscale. The chiralities have been ascribed to the cross-stacking of NRs that can be controlled by clockwise and counter-clockwise rotation. And the simple method is applicable to various inorganic nanorods. This work is interesting and provides a simple and feasible route to adjust assembly form from original arrangement to desired arrangement for somethings needed. I recommend to consider this work for publication after addressing several minor issues as follows.

1. The AFM images in Fig.2l are difficult to correspond SEM image in Fig.3j and optical images in Fig.k. More matched AFM images are required, or at least provide a detailed description on AFM images.

2. CdSe/CdS, TiO₂ and CdSe NRs are coated with dodecanethiol, oleic acid, and tetradecylphosphonic, respectively. Please mention difference from different modifications, just means different surfactants or others. Please mention these in text.

3. Modulating precision to cross-stack nanorods should be discussed with a few words.

4. The analysis of forces suggest to be reflected and this is important for modulating precision. For example, if substrate used for original assembly has an influence on modulating deformation under external force? What is relationships between interaction force between substrate and nanorods, press-and-rotate mechanic force, van der Waals force, electrostatic force, and dissipative force, please give a qualitative analysis at least. Regarding this, please reference the following literatures:

(1) Vortical superlattices in a gold nanorods' self-assembled monolayer, *Nanoscale* 6, 3064, 2014.

(2) Controllable Two-stage droplet evaporation method and its nanoparticle self-assembly mechanism, *Langmuir*, 29, 6232, 2013.

(3) Symmetry control of nanorod superlattice driven by a governing force, *Nature Commun.*, 8:1410, 2017.

Reviewer #3 (Remarks to the Author):

The authors describe an interesting approach to fabricate chiral nanoassemblies by macroscopic grinding. They demonstrated the generality of their approach to a few inorganic Nanomaterials, as well as extend the concept to molecular systems. Overall, the work is technically sound and may be publishable after addressing the following issues:

(1) In Figure 1b, FFT patterns were shown in different regions of TEM images. I wonder the control sample with out the grinding process. It has to be noted that self-assembled structures typically suffer from spatial heterogeneity. 1D building block system has been found to display various possible orientational order even under normal drying-mediated self-assembly process (e.g. *ACS nano* 2016, 10 (1), 967-976). They may need to discuss this in the literature context.

(2) what is the thickness requirement to form the chiral nanossemblies? can they achieve monolayered chiral structures at large scale?

(3) it is expected that grinding may generate heat which may degrade samples or cause structural heterogeneity.

(4) It has also been known that nanorods may have horizontal and vertical configuration during drying-mediated self assembly (e.g. *ACS Nano* 2012, 6 (1), 925-934). I wonder the packing order

can be altered with grinding process. it is perhaps important for them to demonstrate their claim of microscope force to impact on nanoscale ordering and discuss their results with respect to literature context.

Reviewer #4 (Remarks to the Author):

The manuscript submitted by Z. Yang et al described a grinding method to induce chirality in assembled nanorods. As the first step, they obtained assembled film of nanorods by drop-casting. The nanorods stood vertically on the substrate. Then, a rotate pressive mechanical force was applied on the assembled film with another substrate to realize the mechanical grinding. After clockwise and counterclockwise rotation, nanorods in the film lost the transitional ordering and assembled along with the tangent line of the circles, which enabled cross-stacking of the nanorods. These mechanically grinded film showed clear chiral optical (CD and CPL) signal depending on the mechanical rotating direction. This chirality generation approach was demonstrated to be successful in several inorganic nanorods systems. The authors further used the inorganic nanorods chiral solid as template for chirality transfer. The results are very interesting. Inducing chirality in achiral materials is an important research topic. The authors provide a unique approach to generate chirality in assembled nanorod systems. However, the mechanism that facilitate nanoscale chirality by macroscale grinding is yet not clear; how to understand the structural related chiral optical signal of different samples, and how to achieve controllable chiral response was also not discussed in detail in the paper. The paper needs major revision regarding these points before further consideration of publication on Nature Communications. Also I have some questions:

(1) The chirality generation in the assembled nanorods were attributed to the weak van der Waals interaction between the nanorods. In the control experiment for the NRs treated by dithiol, no chiral optical signal was observed after grinding. Does this change if higher pressure was used in the grinding process?

(2) Does the grinding process cause uneven cross-stacking of the nanorod? The degree of cross-stacking of the nanorods should change with the distance (d) between the rods' position and the rotating center. But there was no observed difference of the CPL and CD peak wavelength regarding the different d according to the authors' discussion. Could the authors explain the possible reason?

(3) Does the grinding condition, for instance, pressure, rotating speed, and mechanical grinding processing time, influence the induced cross-stacking of the nanorods? Could the authors provide experimental data for this?

(4) Is it possible to tune the CD/CPL peak position for the chiral nanorod assembly of CdSe-CdS nanorod with a specific aspect ratio, for instance, NR2 (3.5nm diameter, 22.5 nm length)?

Response to Reviewer #1

The outlined manuscript describes a novel, mechanical strategy to achieve structurally chiral assemblies of nanorods (NRs) showing chiroptical properties. The method is based on grinding of solid-state nanorod assemblies, which causes cross-stacking of otherwise parallel NRs. The developed strategy has been tested against different composition of semiconductor NRs (CdSe/CdS, CdSe, TiO₂) and different aspect ratio of NRs, attesting that this methodology provides access to materials exhibiting CD and CPL tuned with the type of NR. Moreover, reversible tuning of chiroptical properties was achieved by adding/removing different organic additives into/from NR assemblies. Claims of the Authors are supported with data from different spectroscopic (UV-Vis, PL, CD, CPL) and microscopic (SEM, TEM, AFM) analyzes. In my opinion this work is truly novel and the proposed methodology well tested using various materials. The described method provides access to circularly polarized luminescent structures which are of great interest for different optoelectronic technologies, and recently attract considerable interest. Thus, I find this publication suitable for publication in Nature communications journal, however only after resolving some issues outlined below.

Reply: We thank the reviewer for reviewing our manuscript and for the positive comments, and hope that our revision (see below) will clarify his/her concerns.

Major comments:

1. In my opinion speaking of CD properties and providing g-factors for NR2 assemblies has limited physical meaning and could be misleading. It seems that after averaging all spectra collected for different rotations (Fig. S4) the CD would be 0.

Reply: Thanks for the comment. In Fig. S4, we rubbed the NRs assemblies in a linear fashion, which is a control experiment in the present study. In fact, the CD would be zero (or very close to zero) after averaging all spectra collected for different rotations in Fig. S4, suggesting that no “genuine CD” effect presents in the NR assemblies when they experience the rubbing in a linear fashion. In order to avoid the misleading, we have removed the term of the CD in Fig S4. Instead, we used the term of observed CD (CD_{OBS}).

2. The CD is measured using a commercial CD spectrometer which is prone to artifacts when thin films are measured, as noted by the Authors. A more advanced technique such as Mueller Matrix Polarimetry or generalized ellipsometry should be used at least on one representative sample to clearly demonstrate this, e.g. A. Mendoza-Galván et al, Mueller matrix spectroscopic ellipsometry study of chiral nanocrystalline cellulose films, 2018 J. Opt. 20 024001; Optical Chirality Determined from Mueller Matrices, Appl. Sci. 2021, 11, 6742. <https://doi.org/10.3390/app11156742>.

Reply: Thanks for the comment. We have read carefully the papers recommended by this reviewer. We agreed with this reviewer that a commercial CD spectrometer may lead to the artifacts when thin films are measured. A more advanced technique such as Mueller Matrix Polarimetry or generalized ellipsometry could distinguish the contribution of the optical signals from CD to LD or LB. However, we found that such equipment is not available to us. Alternatively, we analyzed the contribution of LD to the observed CD (CD_{OBS}) as follows:

$$CD_{OBS} = CD + LD \cos 2\beta \sin \kappa + \frac{1}{6} \left[CDLBD - CDLB^2 + \left(\frac{1}{2} \ln 10 \right)^2 (CD^3 + CDLD^2) \right] \quad (R1)$$

where CB and LB denote the circular and linear birefringences, respectively. The term in square brackets can be ignored because of its negligibly small contribution, and the $\cos 2\beta \sin \kappa$ term is nearly equal to 0.02 for a commercial CD spectropolarimeter equipped with a photoelastic modulator (PEM) (Langmuir 2002, 18, 462). Hence, the formula can be simplified as follows:

$$CD_{OBS} = CD + LD \times 0.02 \quad (R2)$$

Therefore, the contribution of LD in the CD_{OBS} in terms of percentage ($2LD/CD_{OBS}\%$) is $\sim 1.5\%$ for NR2 assemblies subjected to the CW grinding (Supplementary Fig. 7 in Supplementary Information). In sharp contrast, the contribution from the LD effect is markedly greater in the NR2 assemblies experiencing the rubbing process in a linear fashion (Supplementary Fig. 6 in Supplementary Information), and the value is estimated to be over 60%.

We note that the above technique is a preliminary tool to identify the contribution of LD to the observed CD in a commercial spectropolarimeter, and more advanced techniques, such as Mueller Matrix Polarimetry, could identify the optical chirality with better precision (2018 J. Opt. 20 024001; Appl. Sci. 2021, 11, 6742). We have noted this point in the revised manuscript and cited the recommended literature in Ref. 51-52.

We thank again this reviewer for such a fruitful suggestion. We will seek the collaboration with experts in this field in the future.

3. It is not clear to me where 'circularly packed NRs assemblies' are visible in AFM images - I recommend to highlight these assemblies.

Reply: Thanks for the comment. We have performed additional AFM imaging to show the circularly packed NR assemblies, as shown in **Figure R1**. We have included this figure in the Supplementary Information (Fig. 17).

Figure R1. Additional AFM images of the chiral NR2 assemblies after the CW grinding.

4. I'm not convinced that this method could be described as 'mechanochemical'. Maybe mechanical, or 'mechano-physical' (mechanical force gives rise to chiroptical phenomena). If Authors agree with this comment a partial rewriting of the introduction section would be required.

Reply: Thanks for the comment. Currently, the development of mechanochemistry has expanded the scope from the traditional covalent bonding to the non-covalent supramolecular interactions (Chem. Soc. Rev., 2012, 41, 3493). In the present study, although no covalent bonding associated or dissociated (the classical chemical reactions) during the mechanical grinding process, the supramolecular interactions, mainly the aliphatic chain-chain van der Waals interactions between NRs, were remarkably altered after the mechanical grinding processing.

In the revised manuscript, we described our method as the mechanical approach, avoiding the wording of "mechanochemical approach" in the present system.

5. Experimental details should be provided to more precisely describe fabrication of samples doped with dyes. Also, I wonder how true are the simplified schemes showing composites in figure 5. Are the dye molecules actually infiltrating in between NRs? Or rather deposit as aggregates in cracks seen in AFM images?

Reply: In the revised manuscript, we have provided more experimental details on the fabrication of samples in the Supplementary Information as follows:

The substrate for the dye deposition experiments was prepared as-described in the above section, where NR2-CW or NR2-CCW assemblies were on the quartz substrate. To proceed the dye deposition, the dye molecule (Sudan I or Sudan blue II) was dissolved in ethanol with a concentration of 4 mM, followed by the deposition of 50 μ L of the dye solution onto the substrate by a drop casting method. The carrier solvent (ethanol) was removed by evaporation under 50 $^{\circ}$ C for 20 min. Finally, the as-prepared NR-dye composites were characterized by CD/CPL. To proceed the dye deposition/removal cycle, the as-deposited NR/dye composites were immersed into ethanol for 5 min at least three times, followed by the dye deposition (4 mM, 50 μ L).

Concerning on the position of the dye molecules, the optical images of the sample during the dye deposition-removal were recorded under UV light, which showed distinct color changes (**Figure R2a**). It is worth noting that these dyes were homogeneously deposited on the NR assemblies, ruling out that these dyes were only deposited as aggregates in cracks. In fact, these NRs assemblies were hydrophobic in nature, confirmed by the large contact angle of $\sim 97^{\circ}$ (**Figure R2b**). Deposition of less polar dye molecules (such as Sudan I and Sudan blue II) enables their interaction with NRs through solvophobic force, which consequently results in the contact between NRs and dyes at the molecular level. In sharp contrast, deposition of polar dye molecules, such as Rhodamine 6G (R6G), would lead to the phase segregation between NRs and R6G aggregate. As a result, no induced CD signals from R6G could be observed when they were deposited onto the NR2-CW or NR2-CCW substrate (**Figure R2c**).

Figure R2. (a) Optical images of the NR2-CW assemblies with the deposition/removal of Sudan blue II dye, recorded under UV light. (b) Contact angle measurements of the NR2 assemblies and NR2-CW assemblies. (c) CD spectra of NR2/R6G-CW, NR2/R6G-CCW, NR2-CW, and NR2-CCW.

Minor comments:

6. Wavelength at which CD is measured, shown in panels g and h in Fig. 5, should be clearly stated in the figure and/or figure caption.

Reply: Thanks for pointing out this issue. We have provided the wavelength at which CD is measured in panel g and h in Fig. 5.

“(g) CD values at 645 nm of NR2/Sudan blue II-CW after Sudan blue II deposition and removal. (h) CD values at 474 nm of NR2/Sudan I-CW after Sudan I deposition and removal.”

7. All schemes of NR assemblies showcase vertical assemblies, whereas oblique assemblies of NR2 and NR3 give higher g factor values - maybe it would be worth to include at least one scheme for NR2/NR3?

Reply: Thanks for the suggestion. We have included one scheme (**Figure R3**) for the oblique assemblies of NR2 and NR3 for the better understanding for the readers, which is included in the Supplementary Information (Fig. 3a). The scheme is as follows:

Figure R3. Scheme for the grinding process of the NR2 or NR3 assemblies, which showed the oblique assembling behavior.

8. *Circular/linear dichroism are phenomena, not methods, this sentence should be modified: 'The as-processed NR2 assemblies were studied by CD/linear dichroism (LD)'*

Reply: Thanks for the comment. We have modified the as-mentioned sentence as suggested.

“The as-processed NR2 assemblies were studied by a commercial spectropolarimeter (JASCO J-1500) to understand the CD/linear dichroism (LD) effect and intriguingly, both CD and LD signals could be clearly observed from the respective spectra.”

9. *What is the role of silanization of the top quartz substrate used for grinding?*

Reply: Thanks for the comment. we have functionalized the quartz substrate with hydrophobic silane molecules, which were expected to interact with NRs through similar chain-chain van der Waals interactions. Hence, the interaction between NRs and the quartz substrate is mainly from weak van der Waals interaction, which is compatible to that between NRs. When the silanization was not applied to the quartz substrate, the surface hydroxyl group on the quartz could interact with the surface of CdSe/CdS NRs under applied pressure. In this way, the CdSe/CdS NRs may stick to the cover quartz slab, which would destroy the sample.

10. *Authors mention that electronic coupling could be responsible for the emergent chiroptical properties. Thus, maybe it is worth highlighting the difference between photoluminescence of NR samples in dispersions and solid state, confirming the coupling phenomena.*

Reply: Thanks for the comment. Compared to the NRs dispersed in organic solvent, the as-assembled NRs showed a ~8 nm bathochromic shift in the peak position in the photoluminescence spectra (**Figure R4**), revealing that the electronic coupling takes place between adjacent NRs. We have noted this point in the revised manuscript.

This figure is also included in Supplementary Fig. 4.

Figure R4. PL spectra of NR2 in solution and in assembled state.

11. *I wonder if there shouldn't be a difference in CD/CPL dependent on the distance to the center of grinded sample?*

Reply: Thanks for the comment. To further confirm the correlation between the CD intensities and the distance (d) from the rotating center, we have performed multiple CD/CPL experiments at $d = 0$ mm, $d = 2.5$ mm, and $d = 5$ mm, denoted as regions 1, 2, and 3, respectively. The results were shown in **Figure R5** and **R6**. The results showed that both the CD spectra and CPL spectra are very similar at three distinct regions. These results were highly reproducible, as confirmed by five independent grinding experiments. The dimensionless g -factor at 490 nm and g_{lum} at 575 nm of the samples at various regions were ~ 0.009 and ~ 0.05 , respectively. These results confirmed that there was no observed difference of the CPL and CD intensity regarding the different d . Theoretically, the degree of cross-stacking of the NRs should depend on the distance between the NRs' position and the rotating center for a monolayer system. Here, the thickness of the NR is over 1 μm , which is roughly a few tens of layers of the NRs. During the grinding process, the NRs between interlayers could "interlock" each other, which eventually leads to the "equilibrium" structure after 21 s under 1.0 rad s^{-1} .

To further confirm the formation of "equilibrium" structure of NR assemblies, the mechanical grinding processing time varied from 7 to 35 s, which is also suggested by reviewer #4 in Comment #3. The results in **Figure R7** showed that the CD intensity (or g -factor) at 490 nm increases with the increase of the mechanical grinding time at the beginning, which subsequently falls to the steady value after 21 s. These results indicate that the produced chiral NR assemblies are the stable structures, which are probably due to the interlayer locking during the grinding process.

Figure R5. (a) the scheme for the various regions of the NR assemblies for the CD measurements (left) and the g -factor at 490 nm of three different regions from five independent experiments (right); (b-f) CD and absorbance spectra of NR assemblies after grinding processing from five independent experiments.

Figure R6. (a) the scheme for the various regions of the NR assemblies for the CPL measurements (left) and the g_{um} at 575 nm of three different regions from five independent experiments (right); (b-f) CPL and PL spectra of NR assemblies after grinding processing from five independent experiments.

Figure R7. Grinding of NR2 assemblies at various processing times: (a) 7 s; (b) 14 s; (c) 21 s; (d) 28 s; (e) 35 s. (f) The plot of g -factor at 490 nm versus the processing time.

12. It is surprising to me that TEM grid survived grinding - was this method working smoothly or it required several attempts not to damage the carbon film?

Reply: Thanks for the comment. In fact, it is quite struggling for us to prepare the samples for TEM imaging. It requires several attempts not to damage the carbon film. We have noted this information in the revised manuscript

Response to Reviewer #2

This work shows chirality in nanorod assemblies can be generated by using macroscopic grinding, which extends scope of mechanochemistry to macroscale. The chiralities have been ascribed to the cross-stacking of NRs that can be controlled by clockwise and counter-clockwise rotation. And the simple method is applicable to various inorganic nanorods. This work is interesting and provides a simple and feasible route to adjust assembly form from original arrangement to desired arrangement for somethings needed. I recommend to consider this work for publication after addressing several minor issues as follows.

Reply: We thank the referee for reviewing our manuscript and for the positive comments, and hope that our revision (see below) will clarify his/her concerns.

1. *The AFM images in Fig.2l are difficult to correspond SEM image in Fig.3j and optical images in Fig.k. More matched AFM images are required, or at least provide a detailed description on AFM images.*

Reply: We thank this reviewer for pointing out this issue. We have provided additional AFM images in the revised manuscript (**Figure R1**). Since the AFM images and the TEM images show the different length scales, the former one shows the scale of a few micrometers, whereas the latter one shows the scale of tens of nanometers.

Figure R1. Additional AFM images of the chiral NR assemblies after the CW grinding.

2. *CdSe/CdS, TiO₂ and CdSe NRs is coated with dodecanethiol, oleic acid, and tetradecylphosphonic, respectively. Please mention difference from different modifications, just means different surfactants or others. Please mention these in text.*

Reply: Thanks for the comment. We have mentioned the conditions of the nanorods used in the present studies in the revised manuscript.

“In this regard, we have synthesized TiO₂ and CdSe NRs coated with oleic acid and tetradecylphosphonic acid, respectively, and these surface ligands would provide the van der Waals interactions within in the NR assemblies.”

3. *Modulating precision to cross-stack nanorods should be discussed with a few words.*

Reply: Thanks for the comment. The measured CD spectra of the chiral NR assemblies were the signals reflecting an ensemble. However, it is rather difficult to determine the precise configuration of the cross-stacked nanorods. We have noted this point in the revised manuscript.

4. *The analysis of forces suggest to be reflected and this is important for modulating precision. For example, if substrate used for original assembly has an influence on modulating deformation under external force? What is relationships between*

interaction force between substrate and nanorods, press-and-rotate mechanic force, van der Waals force, electrostatic force, and dissipative force, please give a qualitative analysis at least. Regarding this, please reference the following literatures:

(1) Vortical superlattices in a gold nanorods' self-assembled monolayer, *Nanoscale* 6, 3064, 2014.

(2) Controllable Two-stage droplet evaporation method and its nanoparticle self-assembly mechanism, *Langmuir*, 29, 6232, 2013.

(3) Symmetry control of nanorod superlattice driven by a governing force, *Nature Commun.*, 8:1410, 2017.

Reply: Thanks for the comment and suggestion. We have analyzed the forces of the system, and included the recommended literatures in the revised manuscript (Ref. 47-49).

First, we analyzed the intermolecular forces between NRs coated with dodecanethiol. we analyzed the pairwise interparticle energy between two NRs. The model of NRs was built with the diameter and length of 4 and 40 nm, respectively. The surface of NRs was coated with a layer of dodecanethiol with a ligand density of 4 ligands per nm². In the self-assembled NR superlattices, the interaction energy between two neighboring NRs could be from the van der Waals attraction from the CdSe/CdS cores, van der Waals attraction from the aliphatic chains, and the elastic repulsion of the aliphatic chains. The van der Waals attractive energy from the CdSe/CdS cores is mainly determined by the Hamaker constant in hydrocarbon medium. Previous report indicates that the semiconductor core-core van der Waals interaction energy is low, comparable to the thermal energy $k_B T$ (k_B , Boltzmann constant, T , absolute temperature) (J. Am. Chem. Soc. 2007, 129, 15706). In fact, the interaction energy between two NRs of side-by-side packing is mainly from the intermolecular interactions of the surface aliphatic chains. Other noncovalent interactions, such as electrostatic forces or hydrogen bonding, could not be the intermolecular forces between the NRs.

The interaction energy between aliphatic chains on two curved surfaces can be described by the optimal packing model (OPM) from Landman (Faraday Discuss. 2004, 125, 1), which assumes that the ligands lying on the nanocrystal-nanocrystal line pack densely within a narrow volume, as a model to estimate the interaction contribution from the ligand shells. The OPM predicts the interparticle separation d to be

$$d = 2r[(1 + 3\frac{L}{r})^{1/3} - 1] \quad (R1)$$

Where r and L is the radius of the core particle and the length of aliphatic chains (~1.5 nm for dodecanethiol). For the NR of 2 nm in radius, an interparticle separation is calculated to be 1.9 nm, which agrees with our experimental observation from the TEM images. The van der Waals attraction (U_{C12}) between two nearest parallel alkyl chains of length L from N identical basic units ($L = N\lambda$) and separated by a distance D has been given by Salem (J. Chem. Phys. 1962, 37, 2100):

$$U_{C12} = A \frac{3\pi}{8\lambda^2} \frac{L}{D^5} \quad (R2)$$

Where A is the Hamaker constant of methylene units. With Salem's conclusion that the attractive energy is correlated with the length of the alkyl chain, and with the attractive energy for C12 is calculated to be $-9.6 k_B T$ /molecule, we could relate the overlapping length and interaction strength:

$$U_{attr} \approx (-4.8k_B T) \times (2L - d) \quad (R3)$$

The elastic repulsion energy between two C12 chains can be calculated on the basis of the elastic modulus (E), which is on the order of ~0.86 GPa for dodecanethiols (Appl. Phys. Lett. 2009, 94, 131909). Hence the elastic repulsion energy can be estimated to be

$$U_{el} \approx \frac{1}{2} \times \frac{EA_0}{L} \approx (17.2k_B T) \times (2L - d)^2 \quad (R4)$$

Where A_0 is the cross-sectional area of an alkyl chain ($A_0 \approx 0.25 \text{ nm}^2$). Considering the attractive and elastic repulsive energy between two parallel C12 chains, we claim that the overall contribution from the ligand-ligand interactions can be $\sim -1 k_B T$ (The minus indicates the attractive energy). Considering the fact that two NRs are packed parallelly, the interactions from ligands can be described in formula:

$$U_{ligand} \approx N_{lig}[-4.8k_B T \times (2L - d) + (17.2k_B T) \times (2L - d)^2] \quad (R5)$$

Where N_{lig} is the total number of interacted ligands between two NRs. Since the OPM is built on the assumption that the ligands pack densely only with a narrow volume along the contact axis between two nearest neighbors, the N_{lig} depends on the length of NR and the curvature of NR. Considering the high curvature of the NR $\kappa = 0.25 \text{ nm}^{-1}$, we assume that one pair of ligand-ligand interaction is possible at a unit length ($\sim 0.5 \text{ nm}$). Therefore, for the two NRs of side-by-side packing, their attractive energy from the surface ligands is estimated to be $\sim -80k_B T$ ($\sim -3.2 \times 10^{-19} \text{ J}$).

The pressure we loaded on the NR assemblies ranges from 100 to 300 kPa (from 0.1 pN nm^{-2} to 0.3 pN nm^{-2}). For the surface area of the NR is approximately from tens to a few hundred of nm^2 , we estimate that the applied force to each NR is a few tens of pN, which is capable of moving/rotating nanoscale objects. The work is on the order of 10^{-19} J , which is comparable to the attractive energy between two NRs.

Second, in order to investigate how the loaded pressure could impact the grinding process, we have devised a homemade setup shown in **Figure R2a**, which enables us to control the pressure (by various weights) in a precise manner. we loaded different weights to the samples, which were able to generate pressures of 10, 50, 100, 200, and 300 kPa. All the experiments were repeated three times under CW rotations (1.0 rad s^{-1} , 21 s). The results revealed that very low (noisy) CD signals could be produced under 10 kPa. The CD intensity increases with the increase of the loaded pressure, and reaches a maximum value at 200 kPa. Such a trend could be understood by the g -factor value at 490 nm (**Figure R2**), which increases with the increase of the loaded pressure. This result could be interpreted that low pressure could not induce the sufficient shear force that is capable of breaking the van der Waals interactions between NRs.

Third, concerning on the effect of substrate, we have functionalized the quartz substrate with hydrophobic silane molecules, which were expected to interact with NRs through similar chain-chain van der Waals interactions. Hence, the interaction between NRs and the quartz substrate is mainly from weak van der Waals interaction, which is compatible to that between NRs.

Figure R2. (a) Scheme for the homemade device for controlling the applied pressure during the grinding processing of NR assemblies. The material of the device was made of Teflon. The grinding processing of NR assemblies under various pressure: (b) 10 kPa; (c) 50 Pa; (d) 100 kPa; (e) 200 kPa; (f) 300 kPa. (g) The plot of g -factor at 490 nm versus the applied pressure.

Fourth, concerning on the dissipative energy, we applied a low angular velocity of 1 rad s^{-1} for grinding the samples. During grinding the sample, the temperature of the sample was monitored by the electronic contactless thermometer. We did not observe the apparent temperature increase, indicating that the heat generated from the grinding may dissipate to the environment. In other words, the dissipative energy during the grinding process could not be responsible for the symmetry breaking in the NR assemblies.

Response to Reviewer #3

The authors describe an interesting approach to fabricate chiral nanoassemblies by macroscopic grinding. They demonstrated the generality of their approach to a few inorganic Nanomaterials, as well as extend the concept to molecular systems. Overall, the work is technically sound and may be publishable after addressing the following issues:

Reply: We thank this reviewer for the positive comments, and hope that our revision (see below) will clarify his/her concerns.

(1) In Figure 1b, FFT patterns were shown in different regions of TEM images. I wonder the control sample without the grinding process. It has to be noted that self-assembled structures typically suffer from spatial heterogeneity. 1D building block system has been found to display various possible orientational order even under normal drying-mediated self-assembly process (e.g. ACS nano 2016, 10 (1), 967-976). They may need to discuss this in the literature context.

Reply: We thank this reviewer for pointing out this issue. Self-assembly of 1D nanorods under drying-mediated self-assembly may generate several distinct orientations, which probably leads to the spatial heterogeneity of the self-assembled thin film. We have added such discussion into the revised manuscript. The recommended literature has been cited in the revised manuscript (Ref. 44).

(2) what is the thickness requirement to form the chiral nanoassemblies? can they achieve monolayered chiral structures at large scale?

Reply: Thanks for the comment. To understand how the film thickness impacts the chiral nanoassemblies, we prepared the samples differing by the amounts of the nanorods (NR2), which were subjected to the subsequent grinding treatments. After the deposition of the NR2 onto the substrate, the thickness of the film was determined by AFM measurements. Specifically, the as-prepared NR2 film was subjected to the scratching using a tweezer tip, which enables to determine the thickness of the film. Three distinct thicknesses of the NR2 films of ~300, ~540, and ~1100 nm, were obtained under the mass of NR2 of 0.2, 0.4, and 0.8 mg, respectively. After the grinding treatments under identical conditions, distinct CD intensities could be observed for these samples. The intensity of CD signal is markedly greater with the increase of the thickness of the film. When the sample is about 300 nm, rather low intensity of CD signal could be observed. This can also be understood by the *g*-factor at 490 nm. For the NR2 assemblies of 1100 nm in thickness, the *g*-factor could reach 0.01. In sharp contrast, it decreases to ~0.001 for the NR2 assemblies of 300 nm (**Figure R1**).

Figure R1. Optical image (left), AFM image (middle), and the corresponding height profile (right) of NR assemblies of various thicknesses (a) ~300 nm from deposition of 0.2 mg of NR2; (b) ~540 nm from deposition of 0.4 mg of NR2; (a) ~1100 nm from deposition of 0.8 mg of NR2. The absorption (d) and CD (e) spectra of the NR2 assemblies differing by the film thicknesses (mass of NR2) after grinding processing. (f) The plot of g-factor at 490 nm versus the mass of NR2.

(3) it is expected that grinding may generate heat which may degrade samples or cause structural heterogeneity.

Reply: Thanks for the comment. We applied a low angular velocity of 1 rad s^{-1} for grinding the samples. During grinding the sample, the temperature of the sample was monitored by the electronic contactless thermometer. We did not observe the apparent temperature increase, indicating that the heat generated from the grinding may dissipate to the environment.

(4) It has also been known that nanorods may have horizontal and vertical configuration during drying-mediated self assembly (e.g. *Acs Nano* 2012, 6 (1), 925-934). I wonder the packing order can be altered with grinding process. it is perhaps important for them to demonstrate their claim of microscope force to impact on nanoscale ordering and discuss their results with respect to literature context.

Reply: Thanks for the comment and suggestion. It could be of highly interest to understand the effect of the configuration of the NR assemblies on their chiral assembling behavior after the grinding processing. The recommended literature is very interesting and it has been cited in the revised manuscript (Ref. 43). However, the controlling of horizontal and vertical configuration of NR assemblies itself is a challenging topic, which is even more challenging when they are stacked into multiple layers. Up to date, we are unable to control the horizontal and vertical configuration of the NR assemblies. The horizontal assembly of NRs probably requires the surface engineering of the substrate, which in turn enables the contact of the lateral facet of NR with the substrate, leading to the preferential horizontal assembly of NRs.

Response to Reviewer #4

The manuscript submitted by Z. Yang et al described a grinding method to induce chirality in assembled nanorods. As the first step, they obtained assembled film of nanorods by drop-casting. The nanorods stood vertically on the substrate. Then, a rotative pressive mechanical force was applied on the assembled film with another substrate to realize the mechanical grinding. After clockwise and counterclockwise rotation, nanorods in the film lost the transitional ordering and assembled along with the tangent line of the circles, which enabled cross-stacking of the nanorods. These mechanically grinded film showed clear chiral optical (CD and CPL) signal depending on the mechanical rotating direction. This chirality generation approach was demonstrated to be successful in several inorganic nanorods systems. The authors further used the inorganic nanorods chiral solid as template for chirality transfer. The results are very interesting. Inducing chirality in achiral materials is an important research topic. The authors provide a unique approach to generate chirality in assembled nanorod systems.

Reply: We thank this reviewer for the positive comments, and hope that our revision (see below) will clarify his/her concerns.

However, the mechanism that facilitate nanoscale chirality by macroscale grinding is yet not clear; how to understand the structural related chiral optical signal of different samples, and how to achieve controllable chiral response was also not discussed in detail in the paper. The paper needs major revision regarding these points before further consideration of publication on Nature Communications. Also I have some questions:

Reply: Thanks for the comment. First of all, concerning on the mechanism on the nanoscale chirality associated with the macroscale grinding, we believe that the rotational force from the macroscale grinding could break the spatial arrangement of NRs, rendering these NRs chiral. In fact, previous report showed that the handedness of helical supramolecular aggregates formed by achiral molecules can be well controlled by applying rotational forces during the self-assembly process (Nat. Chem. 2012, 4, 201; Nat. Commun. 2018, 9, 2599). To get insight on the chirality generation, we analyzed the pairwise interparticle energy between two NRs. The model of NRs was built with the diameter and length of 4 and 40 nm, respectively. The surface of NRs was coated with a layer of dodecanethiol with a ligand density of 4 ligands per nm² (Determined from ¹H-NMR method, data were not shown). In the self-assembled NR superlattices, the interaction energy between two neighboring NRs could be from the van der Waals attraction from the CdSe/CdS cores, van der Waals attraction from the aliphatic chains, and the elastic repulsion of the aliphatic chains. The van der Waals attractive energy from the CdSe/CdS cores is mainly determined by the Hamaker constant in hydrocarbon medium. Previous report indicates that the semiconductor core-core van der Waals interaction energy is low, comparable to the thermal energy $k_B T$ (k_B , Boltzmann constant, T , absolute temperature) (J. Am. Chem. Soc. 2007, 129, 15706). In fact, the interaction energy between two NRs of side-by-side packing is mainly from the intermolecular interactions of the surface aliphatic chains.

The interaction energy between aliphatic chains on two curved surfaces can be described by the optimal packing model (OPM) from Landman (Faraday Discuss. 2004, 125, 1), which assumes that the ligands lying on the nanocrystal-nanocrystal line pack densely within a narrow volume, as a model to estimate the interaction contribution from the ligand shells. The OPM predicts the interparticle separation d to be

$$d = 2r[(1 + 3\frac{L}{r})^{1/3} - 1] \quad (R1)$$

Where r and L is the radius of the core particle and the length of aliphatic chains (~1.5 nm for dodecanethiol). For the NR of 2 nm in radius, an interparticle separation is calculated to be 1.9 nm, which agrees with our experimental observation from the TEM images. The van der Waals attraction (U_{C12}) between two nearest parallel alkyl chains of length L from N identical basic units ($L = N\lambda$) and separated by a distance D has been given by Salem (J. Chem. Phys. 1962, 37, 2100):

$$U_{C12} = A \frac{3\pi}{8\lambda^2} \frac{L}{D^5} \quad (R2)$$

Where A is the Hamaker constant of methylene units. With Salem's conclusion that the attractive energy is correlated with

the length of the alkyl chain, and with the attractive energy for C12 is calculated to be $-9.6 k_B T$ /molecule, we could relate the overlapping length and interaction strength:

$$U_{attr} \approx (-4.8k_B T) \times (2L - d) \quad (R3)$$

The elastic repulsion energy between two C12 chains can be calculated on the basis of the elastic modulus (E), which is on the order of ~ 0.86 GPa for dodecanethiols (Appl. Phys. Lett. 2009, 94, 131909). Hence the elastic repulsion energy can be estimated to be

$$U_{el} \approx \frac{1}{2} \times \frac{EA_0}{L} \approx (17.2k_B T) \times (2L - d)^2 \quad (R4)$$

Where A_0 is the cross-sectional area of an alkyl chain ($A_0 \approx 0.25 \text{ nm}^2$). Considering the attractive and elastic repulsive energy between two parallel C12 chains, we claim that the overall contribution from the ligand-ligand interactions can be $\sim -1 k_B T$ (The minus indicates the attractive energy). Considering the fact that two NRs are packed parallelly, the interactions from ligands can be described in formula:

$$U_{ligand} \approx N_{lig} [(-4.8k_B T) \times (2L - d) + (17.2k_B T) \times (2L - d)^2] \quad (R5)$$

Where N_{lig} is the total number of interacted ligands between two NRs. Since the OPM is built on the assumption that the ligands pack densely only with a narrow volume along the contact axis between two nearest neighbors, the N_{lig} depends on the length of NR and the curvature of NR. Considering the high curvature of the NR $\kappa = 0.25 \text{ nm}^{-1}$, we assume that one pair of ligand-ligand interaction is possible at a unit length ($\sim 0.5 \text{ nm}$). Therefore, for the two NRs of side-by-side packing, their attractive energy from the surface ligands is estimated to be $\sim -80k_B T$ ($\sim 3.2 \times 10^{-19} \text{ J}$).

The pressure we loaded on the NR assemblies ranges from 100 to 300 kPa (from 0.1 pN nm^{-2} to 0.3 pN nm^{-2}). For the surface area of the NR is approximately from tens to a few hundred of nm^2 , we estimate that the applied force to each NR is a few tens of pN, which is capable of moving/rotating nanoscale objects. The work is on the order of 10^{-19} J , which is comparable to the attractive energy between two NRs.

To further have a better understanding on the interplay between the nanoscale chirality and the macroscale grinding, we have performed additional control experiments, such as which is also suggested by this reviewer in Comment #3. The detailed results were presents in reply to Comment#3.

This additional analysis of the intermolecular interaction was added into Supplementary Information (Note 1) in the revised manuscript.

(1) The chirality generation in the assembled nanorods were attributed to the weak van der Waals interaction between the nanorods. In the control experiment for the NRs treated by dithiol, no chiral optical signal was observed after grinding. Does this change if higher pressure was used in the grinding process?

Reply: Thanks for the comment. In the above response, we have estimated that the attractive energy between two NRs of side-by-side packing is $\sim 3.2 \times 10^{-19} \text{ J}$, which is mainly from the van der Waals interaction of the aliphatic chains. When the surface of NRs is treated by dithiol, the interaction between NRs switches from noncovalent van der Waals interaction to covalent carbon-carbon bonding. Considering the fact that the bond dissociation energy for one C-C bond is approximately $5 \times 10^{-19} \text{ J}$, even greater than the sum of the van der Waals energy between two NRs. When multiple dithiols were linked to the NRs, which is the most probable, which requires more energy to dissociate the C-C bond to render these NRs chiral. On the other hand, the interaction between the NRs and the substrate (quartz) is mainly van der Waals interaction, which is much weaker than that of the covalent interactions between NRs. When high pressure was used in the grinding process, these covalently linked NR assemblies would be detached from the surface of quartz, instead of the moving/rotating action. Therefore, no chiroptical signal was observed after grinding.

(2) Does the grinding process cause uneven cross-stacking of the nanorod? The degree of cross-stacking of the nanorods

should change with the distance (d) between the rods' position and the rotating center. But there was no observed difference of the CPL and CD peak wavelength regarding the different d according to the authors' discussion. Could the authors explain the possible reason?

Reply: Thanks for the comments. The grinding process could cause the uneven cross-stacking of the NRs. To further confirm the correlation between the CD intensities and the distance (d) from the rotating center, we have performed multiple CD/CPL experiments at $d = 0$ mm, $d = 2.5$ mm, and $d = 5$ mm, referred as region 1, 2, and 3, respectively. The results were shown in **Figure R1** and **R2**. The results showed that both the CD spectra and CPL spectra are very similar at three distinct regions. These results were highly reproducible, as confirmed by five independent grinding experiments. The dimensionless g -factor at 490 nm and g_{lum} at 575 nm of the samples at various regions were ~ 0.009 and ~ 0.05 , respectively. These results confirmed that there was no observed difference of the CPL and CD intensity regarding the different d . Theoretically, the degree of cross-stacking of the NRs should depend on the distance between the NRs' position and the rotating center for a monolayer system. Here, the thickness of the NR is over 1 μm , which is roughly a few tens of layers of the NRs. During the grinding process, the NRs between interlayers could "interlock" each other, which eventually leads to the "equilibrium" structure after 21 s under 1.0 rad s^{-1} .

To further confirm the formation of "equilibrium" structure of NR assemblies, the mechanical grinding processing time varied from 7 to 35 s, which is also suggested by this reviewer in Comment #3. The results in **Figure R3** showed that the CD intensity (or g -factor) at 490 nm increases with the increase of the mechanical grinding time at the beginning, which subsequently falls to the steady value after 21 s. These results indicate that the produced chiral NR assemblies are the stable structures, which are probably due to the interlayer locking during the grinding process.

Figure R1. (a) the scheme for the various regions of the NR assemblies for the CD measurements (left) and the g -factor at 490 nm of three different regions from five independent experiments (right); (b-f) CD and absorbance spectra of NR assemblies after grinding processing from five independent experiments.

Figure R2. (a) the scheme for the various regions of the NR assemblies for the CPL measurements (left) and the g_{lum} at 575 nm of three different regions from five independent experiments (right); (b-f) CPL and PL spectra of NR assemblies after grinding processing from five independent experiments.

Figure R3. Grinding of NR2 assemblies at various processing times: (a) 7 s; (b) 14 s; (c) 21 s; (d) 28 s; (e) 35 s. (f) The plot of g -factor at 490 nm versus the processing time.

(3) Does the grinding condition, for instance, pressure, rotating speed, and mechanical grinding processing time, influence the induced cross-stacking of the nanorods? Could the authors provide experimental data for this?

Reply: Thanks for the suggestion. In the revised manuscript, we have provided the additional experiments, such as the effect of pressure, rotating speed, and the mechanical grinding processing time on the CD spectra of the NR assemblies. In order to investigate the various grinding conditions, we have devised a homemade setup shown in **Figure R4a**, which enables us to control the pressure (by various weights), the rotating speed, as well as the mechanical grinding processing time in a precise

manner. In the first set of experiments, we loaded different weights to the samples, which were able to generate pressures of 10, 50, 100, 200, and 300 kPa. All the experiments were repeated three times under CW rotations (1.0 rad s^{-1} , 21 s). The results revealed that very low (noisy) CD signals could be produced under 10 kPa. The CD intensity increases with the increase of the loaded pressure, and reaches a maximum value at 200 kPa. Such a trend could be understood by the g -factor value at 490 nm (**Figure R4**), which increases with the increase of the loaded pressure. This result could be interpreted that the low pressure could not induce the sufficient shear force that is capable of breaking the van der Waals interactions between NRs.

Another control experiment has been done by keeping the pressure constant (200 kPa), differing by the rotating speed, in terms of angular velocity at 0.1, 0.25, 0.5, 1.0, and 1.5 rad s^{-1} . Likewise, all the experiments were repeated three times under CW rotations for 21 s, and the results were presented in **Figure R5**. We found that CD intensities of the NR assemblies are highly associated with the rotation speed. With the increase of the of the rotation speed, the CD intensity increases, which could also be observed in the g -factor at 490 nm. This result is interesting and is unexpected. In the fluid system, the shear stress increases with the increase of shear rate (rotation speed). However, here the NR assemblies were solids at room temperature, which also showed the “fluid-like” behavior—the increase of the rotation speed (shear rate) leads to the greater shear stress, which in turn results in the higher g -factor of the NR assemblies. The “fluid-like” behavior of the NR assemblies could be associated with their surface aliphatic chains, which equip these NRs with very low intermolecular attractive energy. Nevertheless, the “fluid-like” behavior of the NR solids (or nanoparticle solids) needs further investigations in the future studies.

Lastly, we performed the control experiments by tuning the grinding processing time, while keeping the other conditions constant (pressure of 200 kPa, rotation speed 1.0 rad s^{-1} , CW direction). The experiments were also repeated three times, and the results were shown in **Figure R3**. It could be found that a short period of the grinding (7 s) could not lead to the strong CD signals of the NR assemblies, which increases with the increase of the processing time. This can be understood that a short period of rotation could not efficiently drive the breaking of the attractive van der Waals interactions between NRs. After 21 s, strong CD signal could be observed. It is worth noting here that further prolonging the processing time ($> 21 \text{ s}$) could not result in stronger CD signal, indicating that these NR assemblies were in the stable state, which is probably from the interlayer locking of NRs after the grinding processing.

We have included these results and Figures into the Supplementary information in the revised manuscript. We sincerely thank this reviewer raise these critical comments.

Figure R4. (a) Scheme for the homemade device for controlling the applied pressure during the grinding processing of NR assemblies. The material of the device was made of Teflon. The grinding processing of NR assemblies under various pressure: (b) 10 kPa; (c) 50 Pa; (d) 100 kPa; (e) 200 kPa; (f) 300 kPa. (g) The plot of g -factor at 490 nm versus the applied pressure.

Figure R5. Control experiments carried out at various rotating speed: (a) 0.1 rad s⁻¹; (b) 0.25 rad s⁻¹; (c) 0.5 rad s⁻¹; (d) 1.0 rad s⁻¹; (e) 1.5 rad s⁻¹; (f) The plot of g-factor at 490 nm versus the angular velocity.

(4) Is it possible to tune the CD/CPL peak position for the chiral nanorod assembly of CdSe-CdS nanorod with a specific aspect ratio, for instance, NR2 (3.5nm diameter, 22.5 nm length)?

Reply: Thanks for the comment. In the current studies, we found that the CD/CPL peak position for the chiral NR assemblies could not be effectively tuned. The additional experiments we have done in the reply to the Comment #3 suggested that the CD/CPL peak positions of the CdSe/CdS NR assemblies could not be effectively tuned by varying the grinding processing conditions.

Reviewer comments, second round review

Reviewer #1 (Remarks to the Author):

I would like to thank the Authors for providing detailed responses to my comments, including new data, and not overstating their work (e.g. changing mechanochemical to mechanical). In my opinion, this work is now suitable for publication in Nature Communications journal.

Reviewer #2 (Remarks to the Author):

The authors have revised the manuscript resulting in a promotion in understanding, readability, scientific quality, and so on. The revised version has addressed all my concerns and thus I agree to accept this work for publication in Nature Commum.

Reviewer #3 (Remarks to the Author):

The authors have addressed all my comments well. It is now publishable

Reviewer #4 (Remarks to the Author):

Thanks for the authors' reply. My concerns and questions have been solved. I think the manuscript is ready for publication in Nature Communications now.

Reviewer #1 (Remarks to the Author):

I would like to thank the Authors for providing detailed responses to my comments, including new data, and not overstating their work (e.g. changing mechanochemical to mechanical). In my opinion, this work is now suitable for publication in Nature Communications journal.

Response: We thank the reviewers for their valuable comments and we are glad that the reviewers were satisfied with our revisions.

Reviewer #2 (Remarks to the Author):

The authors have revised the manuscript resulting in a promotion in understanding, readability, scientific quality, and so on. The revised version has addressed all my concerns and thus I agree to accept this work for publication in Nature Commum.

Response: We thank the reviewers for their valuable comments and we are glad that the reviewers were satisfied with our revisions.

Reviewer #3 (Remarks to the Author):

The authors have addressed all my comments well. It is now publishable

Response: We thank the reviewers for their valuable comments and we are glad that the reviewers were satisfied with our revisions.

Reviewer #4 (Remarks to the Author):

Thanks for the authors' reply. My concerns and questions have been solved. I think the manuscript is ready for publication in Nature Communications now.

Response: We thank the reviewers for their valuable comments and we are glad that the reviewers were satisfied with our revisions.